# CBGTPy: An extensible cortico-basal ganglia-thalamic framework for modeling biological decision making

Matthew Clapp[1], Jyotika Bahuguna[1], Cristina Giossi[2,3], Jonathan E. Rubin[4,5]*, Timothy Verstynen[1,4]*, Catalina Vich[2,3]*

1 Department of Psychology & Neuroscience Institute, Carnegie Mellon University, Pittsburgh, Pennsylvania, United States of America, 2 Departament de Ciències Matemàtiques i Informàtica, Universitat de les Illes Balears, Palma, Spain, 3 Institute of Applied Computing and Community Code, Palma, Spain, 4 Center for the Neural Basis of Cognition, Pittsburgh, Pennsylvania, United States of America, 5 Department of Mathematics, University of Pittsburgh, Pittsburgh, Pennsylvania, United States of America

☉ These authors contributed equally to this work.
* jonrubin@pitt.edu (JER); timothyv@andrew.cmu.edu (TV); catalina.vich@uib.es (CV)

## Abstract

Here we introduce CBGTPy, a virtual environment for designing and testing goal-directed agents with internal dynamics that are modeled on the cortico-basal-ganglia-thalamic (CBGT) pathways in the mammalian brain. CBGTPy enables researchers to investigate the internal dynamics of the CBGT system during a variety of tasks, allowing for the formation of testable predictions about animal behavior and neural activity. The framework has been designed around the principle of flexibility, such that many experimental parameters in a decision making paradigm can be easily defined and modified. Here we demonstrate the capabilities of CBGTPy across a range of single and multi-choice tasks, highlighting the ease of set up and the biologically realistic behavior that it produces. We show that CBGTPy is extensible enough to apply to a range of experimental protocols and to allow for the implementation of model extensions with minimal developmental effort.

## 1 Introduction

With the rise of fields like cognitive computational neuroscience [1], there has been a resurgence of interest in building biologically realistic models of neural systems that capture prior observations of biological substrates and generate novel predictions at the cellular, systems, and cognitive levels. In many cases, researchers rely on off-the-shelf machine learning models that use abstracted approximations of biological systems (e.g., rate-based activity and rectified linear unit gating, among others) to simulate properties of neural circuits [2–4]. For researchers interested primarily in cortical sensory pathways, these systems work well enough at making behavioral and macroscopic network predictions [5], but they often fail to provide biologically realistic predictions about underlying cellular dynamics that can be tested *in vivo*. Although there are a wealth of biologically realistic simulations of cortical and non-cortical pathways that have helped to significantly advance our understanding of BG function, these

**Data Availability Statement:** All code source files are available publicly on Github (https://github.com/CoAxLab/CBGTPy).

**Funding:** MC, JB, TV and JER are partly supported by National Institutes of Health (https://www.nih.gov) awards R01DA053014 and R01DA059993 as part of the CRCNS program. CG and CV are supported by the PCI2020-112026 project, and CV is also supported by the PCI2023-145982-2, both funded by MCIN/AEI/10.13039/501100011033 (https://www.ciencia.gob.es/site/MICINN/aei) and by the European Union "NextGenerationEU"/PRTR (https://next-generation-eu.europa.eu/) as part of the CRCNS program. CG is also supported by the Conselleria de Fons Europeus, Universitat i Cultura del Govern de les Illes Balears (https://www.caib.es/sites/participacio/ca/l/conselleria_de_fons_europeus_universitat_i_cultura/) under grant FPU2023-008-B. The authors assert that these sponsors did not influence the study design, data collection and analysis, decision to publish, or preparation of the manuscript.

**Competing interests:** The authors have declared that no competing interests exist.

are often designed to address very narrow behaviors and lack flexibility for testing predictions across multiple experimental contexts [6–10].

Here we present a scientifically-oriented tool for creating model systems that emulate the control of information streams during decision making in mammalian brains. Specifically, our approach mimics how cortico-basal ganglia-thalamic (CBGT) networks are hypothesized to regulate the evidence accumulation process as agents evaluate response options. The goal of this tool, called CBGTPy, is to provide a simple and easy-to-use spiking neural network simulator that reproduces the structural and functional properties of CBGT circuits in a variety of experimental environments. The core aim of CBGTPy is to enable researchers to derive neurophysiologically-realistic predictions about basal ganglia dynamics under hypothesis-driven manipulations of experimental conditions.

A key advantage of our CBGTPy framework is that it separates most properties of the behaving agent from the parameters of the environment, such that experimental parameters can be tuned independently of the agent properties and vice versa. We explicitly distinguish the agent (Section 2.3.1) from the environment (Section 2.3.2). The agent generates two behavioral features—action choice and decision time—that match the behavioral data typically collected in relevant experiments and affords users the opportunity to analyze the simultaneous activity of all CBGT nuclei under experimental conditions. The flexibility of the environment component in CBGTPy allows for the simulation of both simple and complex experimental paradigms, including learning tasks with complex feedback properties, such as volatility in action-outcome contingencies and probabilistic reward scenarios, as well as rapid action control tasks (e.g., the stop signal task). On the biological side, CBGTPy incorporates biologically-based aspects of the underlying network pathways and dynamics, a dopamine-dependent plasticity rule [10], and the capacity to mimic targeted stimulation of specific CBGT nuclei (e.g., optogenetic stimulation). CBGTPy also allows the easy addition of novel pathways, as well as modification of network and synaptic parameters, so as to enable modeling new developments in the CBGT anatomy as they emerge in the literature. After a brief review of the CBGT pathways in the next subsection, in Section 2 we provide a full description of the structure, use, and input parameters of CBGTPy. In Section 3, we go on to present examples of its usage on a variety of standard cognitive tasks, before turning to a discussion in Section 4. Various appendices (S1, S2, S4 and S5 Appendices) present additional details about the CBGT model and CBGTPy toolbox, including the implementation of synaptic plasticity and a guide for CBGTPy installation.

Recent findings have suggested that the simple concepts of rigidly parallel feedforward basal ganglia (BG) pathways may be outdated [11, 12] (We use traditional terminology of "direct" and "indirect" pathways and SPNs (e.g, Fig 1). While we recognize that the idea of a unified indirect pathway is outdated, it is useful to maintain a term to refer to the complement of the direct projection from dSPNs to GPi and the ascending pallidostriatal connections.), and part of the motivation for CBGTPy is to provide a tool for developing and exploring more nuanced, updated theories of CBGT dynamics as new discoveries are made. Indeed, achieving a full understanding of CBGT circuit-level computations requires the development of theoretical models that can adapt with and complement the rapidly expanding empirical evidence on CBGT pathways. The fundamental goal of the CBGTPy toolbox is to provide a framework for this rapid theoretical development, which balances biological realism with computational flexibility and extensibility.

## 2 The toolbox

The core of the CBGTPy toolbox comprises an implementation of a spiking model CBGT network tuned to match known neuronal firing rates and connection patterns that have been

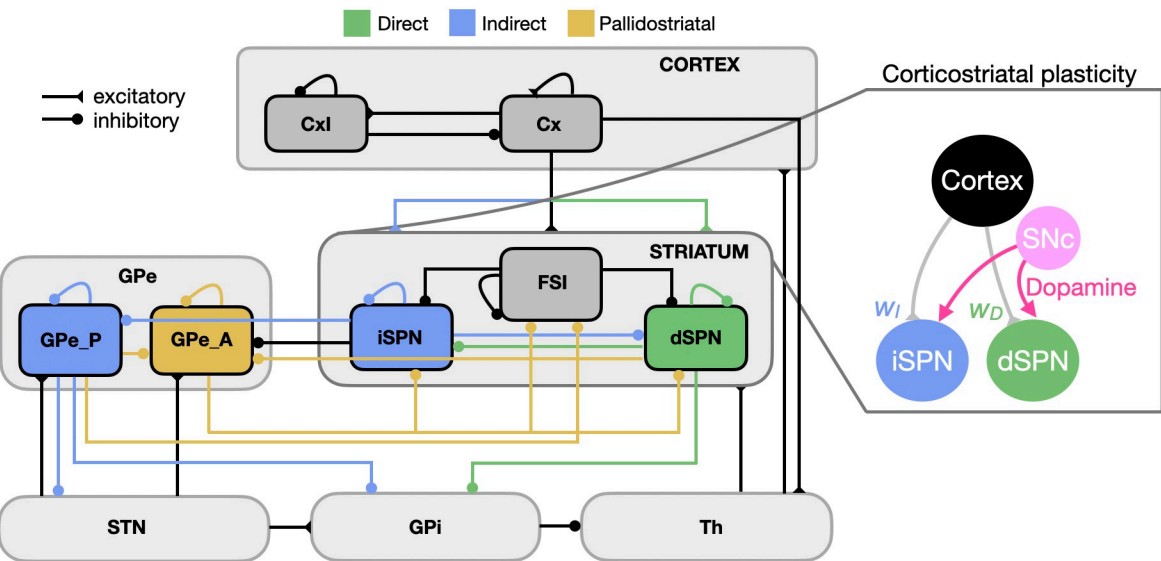

**Fig 1. Overview of the CBGT network.** The connectivity and cellular components of each CBGT channel in a CBGTPy agent are based on known biology. Direct pathway connections are shown in green, indirect pathway connections are shown in blue, and pallidostriatal connections are represented in gold, with arrows ending in circles marking the postsynaptic sites of inhibitory connections and those ending in triangles for excitatory connections. Dopaminergic feedback signals associated with rewards following actions induce plastic changes in corticostriatal synapses (pink arrows). A trial begins when the Cx population receives a stimulus and the model assumes a decision is made when the activity of the Th population reaches a certain threshold. In the current implementation of the network, the pallidostriatal pathways (gold colors) are only considered for the stop signal task.

previously used to study various aspects of basal ganglia function in cognitive tasks [13–16]. The CBGT network model is composed of 6 different regions/nuclei shown in Fig 1: a cortical component, segregated into excitatory (Cx) and inhibitory (CxI) subpopulations; striatum, containing two subpopulations of spiny projection neuron (dSPNs involved in the so-called direct pathway, and iSPNs, involved in the indirect pathway) and also fast-spiking interneurons (FSI); external globus pallidus (GPe), which is divided into prototypical (GPeP) and arkypallidal (GPeA) subpopulations; subthalamic nucleus (STN); internal segment of globus pallidus (GPi); and a pallidal-receiving thalamic component (Th), which receives input from GPi and Cx and projects to cortical and striatal units.

Within each region, we model a collection of spiking point neurons, modeled in a variant of the integrate-and-fire framework [17] to include the spiking needed for synaptic plasticity while still maintaining computational efficiency. Numerical integration is performed via custom Cython code, rather than relying on existing frameworks, such as NEURON [18], BRIAN [19], or NetPyNE [20], a design choice which simplified the overall software stack. The core strengths of these frameworks are in the simulation of multi-scale or multi-compartment models, whereas one of the strengths of the CBGTPy model is the high level of direct control that can be exerted over the neural parameters throughout the interactions between the network and its environment (see Section 2.1). The integration is performed in a partially-vectorized manner, in which each variable is represented as a list of Numpy arrays, one array per neural population. Further details of the implementation of this network, including all relevant equations and parameter values, are provided in S1 Appendix.

CBGTPy allows for the simulation of two general types of tasks that cover a variety of behavioral experiments used in neuroscience research. The first of these tasks is a discrete decision-making paradigm (*n-choice task*) in which the activity of the CBGT network results in the

selection of one choice among a set of options (see Section 3.1). If plasticity is turned on during simulations, phasic dopamine, reflecting a reward prediction error, is released at the corticostriatal synapses and can modify their efficacy, biasing future decisions. We note that the inclusion of a biologically-realistic, dopamine-based learning mechanism, in contrast to the error gradient and backpropagation schemes present in standard artificial agents, represents an important feature of the model in CBGTPy. We present the details of this learning mechanism in S2 Appendix.

The second of these tasks is a stop signal paradigm (*stop signal task*), where the network must control the execution or suppression of an action, following the onset of an imperative cue (see Section 3.2). Here activity of the indirect and pallidostriatal pathways, along with simulated hyperdirect pathway control, determines whether a decision is made within a pre-specified time window. The probability of stop and the relevant RT distributions can be recorded across different values of parameters related to the stop signal.

## 2.1 Agent-environment paradigm

We have adopted an environment-agent implementation architecture, where the internal properties of the CBGT network (the agent) are largely separated from the external properties of the experiment (the environment). Interaction between the agent and environment is limited in scope, as shown in Fig 2, and occurs only at key time points in the model simulation. The core functionality of the agent is the mapping from stimuli to decisions and the implementation of post-decision changes (e.g., synaptic strength updates) to the CBGT network, while the environment serves to present stimuli, cues, and rewards.

CBGTPy uses a data-flow programming paradigm, in which the specification of computing steps is separated from the execution of those steps [21]. Internally, the initialization and simulation of the agent-environment system is divided into a large number of specialized functions, each addressing specific tasks. These code blocks are then organized into sequences, referred to as pipelines. Only after a pipeline is constructed is it executed, transforming any input data into output data. The use of pipelines allows for individual code blocks to be rearranged, reused, and modified as necessary, leading to efficient code reuse.

One of the main benefits of the data-flow design is its synergy with the Ray multiprocessing library for Python. Ray operates on a client-server model and allows for the easy distribution of tasks and worker processes based on the available resources [22]. While the sequence of steps for running a simulation can be constructed locally, those same steps can be distributed and performed remotely on the Ray server. As a result, CBGTPy directly supports running on any system that can support a Ray server, which includes high-core-count computing clusters, while maintaining the exact same interface and ease-of-use as running simulations on a local machine. The user, however, can choose to run the model without any multiprocessing library or an alternative multiprocessing library to Ray. These options are explained in detail in the Section 2.2.

In the following subsections, we explain all the details of the toolbox by separately describing the agent and environmental components that can be changed by the user. The CBGTPy toolbox can be found in the Github repository https://github.com/CoAxLab/CBGTPy/tree/main. The instructions to install it and the list of functions contained in the toolbox can be found in S3 and S4 Appendices, respectively.

## 2.2 Setting up a simulation

One of the objectives of CBGTPy is to enable end users to easily run simulations with default experimental setups. Furthermore, users can specify parameter adjustments with minimal

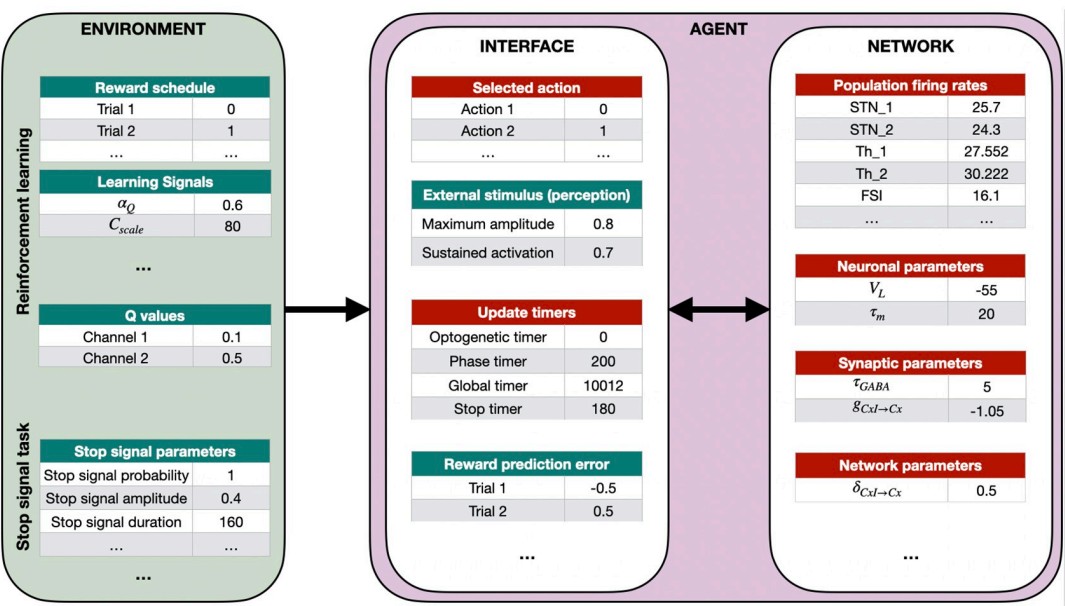

**Fig 2. Example of interactions between the environment and the agent (CBGT network).** The segregation of tasks between the environment and agent allows for independent modification of both. Tables with green title box depict some of the variables that can be easily modified by the user (see Section 2.3) while those in red are automatically updated or set internally. We have divided these variables into two sections: agent-related (Section 2.3.1) and environment-related (Section 2.3.2). The arrow from the environment to the agent block is unidirectional because once the stimulation starts it is not possible to change the environment. The arrow between the interface and network is bidirectional because they are always in constant interaction with each other, the interface controlling the simulation while the actual agent evolves the CBGT network.

effort, specifically through use of a `configuration` variable, which we describe in greater detail in the following sections.

The following list contains a mandatory set of instructions to be executed in order to implement the entire process associated with running a simulation. These instructions will proceed with a default set of parameters. We also provide two example notebooks (n-choice task and stop signal task), which can be found in the repository and include these steps and commands.

- Import all relevant functions.

- Create the main pipeline.

- Import the relevant `paramfile` for the selected experiment type.

- Create `configuration` dictionary with default values.

- Run the simulation, specifying which multiprocessing library to use.

- Extract relevant data frames (e.g., firing rates, reaction times, performance).

- Save variables of interest as pickle files.

- Plot variables of interest (e.g., firing rates and reward data frames).

We explain each of these steps in detail. Note that if Ray multiprocessing is being used, changing the local IP node (e.g., when the underlying Wi-Fi/LAN network has changed) requires stopping the previous instance of the Ray server and restarting it with the newly assigned IP. We also explain how to shut down the Ray server at the end of this section.

**2.2.1 Import relevant functions.** All the relevant imports can be implemented with the following commands:

```
import pandas as pd
import numpy as np
import cbgt as cbgt
import pipeline_creation as pl_creat
import plotting_functions as plt_func
import plotting_helper_functions as plt_help
import postprocessing_helpers as post_help
```

**2.2.2 Create the main pipeline.** Here the user can choose to run either the n-choice task or the stop signal task by assigning a variable `experiment_choice`. (If the variable `experiment_choice` is not set, the pipeline creation gives an error because of the ambiguity of which experiment to run.). Depending on the choice of this variable a relevant pipeline is created. A pipeline consists of all the modules required to run a task and returns a pipeline object that can be used.

For a basic n-choice task, the `experiment_choice` needs to be set as

```
experiment_choice = "n-choice"
```

while for a basic stop signal task, it is set as

```
experiment_choice = "stop-signal"
```

In both cases, the user also should decide how many choices or action channels the current instance of CBGTPy should create and run, using the variable `number_of_choices`:

```
number_of_choices = 2
```

While a common version of this decision making task is run with number_of_choices = 2, it can also be run for an arbitrary number of choices (i.e., number_of_choices $\geq$ 1). The change in the number of choices and corresponding action channels requires scaling of some of the parameters in order to ensure maintenance of the same amount of input to certain

shared CBGT nuclei irrespective of the number of action channels. We explain these scaling schemata in S5 Appendix. Please note that some parameters have to be appropriately updated in the configuration variables (e.g., Q data frame, channel names) according to the `number_of_choices` selected. We explicitly mention which parameters should be updated with the number of choices as we describe them below, and we include example notebooks in the repository for different cases.

The pipeline is created with commands

```
pl_creat.choose_pipeline(experiment_choice)
pl = pl_creat.create_main_pipeline(runloop=True)
```

### 2.2.3 Import the relevant `paramfile` for the selected experiment type.
The `paramfile` contains dictionaries of default parameter values for the neural populations and plasticity model based on the choice of the experiment.

```
if experiment_choice == 'stop-signal':
  import stopsignal.paramfile_stopsignal as paramfile
elif experiment_choice == 'n-choice':
  import nchoice.paramfile_nchoice as paramfile
```

The imported attributes, which can be listed out using `dir(paramfile)`, can be modified according to the user's preferences. For example, setting the cellular capacitance value to 0.5 is accomplished with

```
paramfile.celldefaults['C'] = 0.5
```

### 2.2.4 Create `configuration` dictionary with default values.
The `configuration` variable is a dictionary in which some parameters take internally set default values, whereas others need to be assigned values to run a simulation. A minimal `configuration` variable consists of the following parameters, the details of which are described in S2–S7 Tables and explained separately in Section 2.3.

```
configuration = {
  "experimentchoice": experiment_choice,
  "seed": 0,
  "inter_trial_interval": None, # default = 600ms
```

```
    "thalamic_threshold": None, # default 30sp/s

    "movement_time": None, # default sampled from N(250,1.5)

    "choice_timeout": None, # default 1000

    "params": paramfile.celldefaults,

    "pops": paramfile.popspecific,

    "receps": paramfile.receptordefaults,

    "base": paramfile.basestim,

    "dpmns": paramfile.dpmndefaults,

    "dSPN_params": paramfile.dSPNdefaults,

    "iSPN_params": paramfile.iSPNdefaults,

    "channels": pd.DataFrame([["left"], ["right"]], columns=
["action"]),

    "number_of_choices": number_of_choices,

    "newpathways": None,

    "Q_support_params": None,

    "Q_df": None,

    "n_trials": 3,

    "volatility": [1,"exact"],

    "conflict": (1.0, 0.0),

    "reward_mu": 1,

    "reward_std": 0.1,

    "maxstim": 0.8,

    "corticostriatal_plasticity_present":True,

    "record_variables": ["weight", "optogenetic_input"],

    "opt_signal_present": [True],

    "opt_signal_probability": [[1]],

    "opt_signal_amplitude": [0.1],

    "opt_signal_onset": [20.],

    "opt_signal_duration": [1000.],

    "opt_signal_channel": ["all"],

    "opt_signal_population": ["dSPN"],

    "sustainedfraction": 0.7
}
```

Note that the parameter `corticostriatal_plasticity_present` does not have to be introduced in the configuration dictionary when running the stop signal task (see reference on the example notebook). Additionally, it is important to include the stop signal parameters within the `configuration` dictionary when executing the stop signal task.

```
configuration = {
  "stop_signal_present": [True, True],
  "stop_signal_probability": [1., 1.],
  "stop_signal_amplitude": [0.6, 0.6],
  "stop_signal_onset": [60.,60.],
  "stop_signal_duration": ["phase 0",165.],
  "stop_signal_channel": ["all","left"],
  "stop_signal_population":["STN","GPeA"],
}
```

More details will be provided in the corresponding sections. For reference, you can find an example notebook on github.

**2.2.5 Run the simulation.** At this stage, the user can choose the number of cores to be used (num_cores) and the number of parallel simulations that should be executed with the same `configuration` variable but a different random seed (num_sims). Moreover, the user can optionally specify one of the two supported multiprocessing libraries, Ray and Pathos, to use to run the simulation. Ray is a library providing a compute layer for parallel processing. To start the Ray server, on the command line, run Ray server to execute the head node and obtain the local IP node, in the following way:

```
ray start --head --port=6379 --redis-password="cbgt"
```

This command should list a local IP along with a port number. Hence, to initiate a Ray client that connects to the server started above, the user will have to substitute the local IP node, obtained from the previous command line, in place of `<local ip node>` in

```
ray start --address=<local ip node>:6379 --redis-password="cbgt"
```

Any port number that is free to use in the machine can be used. Here port number 6379 is used, which is the default port number for Ray. To use Ray, the last step consists of setting the variable `use_library` in the notebook to 'ray'. As an alternative to Ray, Pathos is a library that distributes processing across multiple nodes and provides other convenient improvements over Python's built-in tools. To use Pathos, no additional setup is required beyond setting the variable `use_library` to 'pathos'. If the user does not want to use any of the above-mentioned libraries, this should be specified by setting the variable `use_library` to 'none'. The simulation is performed by filling in values for these variables in the following command and executing it:

```
results = cbgt.ExecutionManager

(cores=num_cores,use=use_library).run([pl]*num_sims,[con-
figuration]*num_sims))
```

To ensure the simulation results are both reproducible and robust to minor changes in initial conditions, CBGTPy offers control over the pseudorandom number generator seed. The random seed controls the initial conditions of the network, including precisely which neurons connect together, under the constraint of the given connection probabilities and the baseline background activity levels of the CBGT nuclei. The simulation returns a `results` object, which is a dictionary containing all the data produced by the model, typically organized into data frames [23].

**2.2.6 Extract relevant data frames.** Once the `results` object has been returned, specific variables and tables of interest can be extracted (see S1 Fig). All the variables available can be listed by accessing the keys of the variable `results`, which is done by executing the following command:

```
results[0].keys()
```

All environmental variables passed to the simulation can also be accessed here for cross-checking.

Some additional data frames related to the simulation are also returned. One of these data frames is `results[0]["popfreqs"]`, which returns the population firing rate traces of all nuclei, with each neuronal subpopulation as a column and each time bin of simulated time as a row (see S2 Fig). This data frame can be addressed directly by executing its name in a command line.

Another relevant data frame is `datatables[0]`, which contains a list of chosen actions, optimal actions, reward outcomes, and decision times for all of the trials in the simulation (see S3 Fig). When running multiple simulations in parallel (i.e., `num_sims` > 1), `datatables[i]` is returned, where *i* indicates the corresponding thread. This data frame can be extracted by first executing the command

```
datatables = cbgt.collateVariable(results,"datatables")
```

and next typing `datatables[i]` on the command line, to access the results of the $i^{th}$ simulation.

As part of the model's tuning of dopamine release and associated dopamine-dependent cortico-striatal synaptic plasticity, the model maintains Q-values for each action, updated according to the Q-learning rule (details in S2 Appendix). These values are available in `results["Q_df"]`, where each column corresponds to one of the possible choices (see S4 Fig). These data frames are designed for easy interpretation and use in later data processing steps. It should be however noted that while Q-values are updated and maintained by CBGTPy, the q-values do not influence the selection of decision choice. The selection of decision choice is solely dependent on the corticostriatal weights.

CBGTPy also provides a function to extract some of these data frames in a more processed form. The specific command for the n-choice task is given by

```
firing_rates, reward_q_df, performance, rt_dist, total_per-
formance =

plt_help.extract_relevant_frames(results, seed,
experiment_choice)
```

where `firing_rates` provides a stacked up (pandas command `melt`) version of the `results[0]["popfreqs"]` that can be used in the *seaborn.catplot()* plotting function, `reward_q_df` compiles data frames for reward and q-values, `performance` returns the percentage of each decision choice, `rt_dist` returns the reaction time distribution for the simulation, and `total_performance` compares the `decision` and `correctdecision` in `datatables[i]` and calculates the performance of the agent. Depending on the experiment choice, the function returns relevant data frames. Note that for the stop signal task, the data frames returned are just `firing_rates` and `rt_dist`.

The time-dependent values of the recorded variables can also be extracted for both the n-choice task and the stop signal task using the following command:

```
recorded_variables =

post_help.extract_recording_variables(results,

  results[0]["record_variables"],

  seed)
```

Presently, for the n-choice task, CBGTPy only allows recording the variable `weight` or `optogenetic_input`. The former can be used to track the evolution of corticostriatal

weights during a n-choice experiment. The variable `optogenetic_input` can be recorded and plotted to check if the optogenetic input was applied as intended to the target nuclei. The list of variables to be recorded should be specified in the configuration variable. In the example of the configuration variable used above, both `weight` and `optogenetic_input` are recorded.:

```
configuration = {

..

"record_variables": ["weight", "optogenetic_input"],

..

}
```

An example of plotting these data frames is included in the example python notebook in the GitHub repository. In addition, for the stop signal task, CBGTPy also allows the recording of the variable "`stop_input`", which can be used to check if the stop signal inputs were applied correctly to the target nuclei.

**2.2.7 Save variables of interest as pickle files.** All the relevant variables can be compiled together and saved in a single pickle file. Pickle files provide a method for saving complex Python data structures in a compact, binary format. The following command saves the object `results` with additional data frames of `popfreqs` and `popdata` into a pickle file `network_data` in the current directory:

```
cbgt.saveResults(results, "network_data",
["popfreqs","popdata"])
```

**2.2.8 Basic plotting functions (plot firing rates and reward data frame).** CBGTPy provides some basic plotting functions. The `firing_rates` data frame from the above functions can be passed to function `plot_fr`, which returns a figure handle that can be saved (see Fig 3), as follows:

```
FR_fig_handles = plt_func.plot_fr(firing_rates, datatables,
results,
  experiment_choice, display_stim)
```

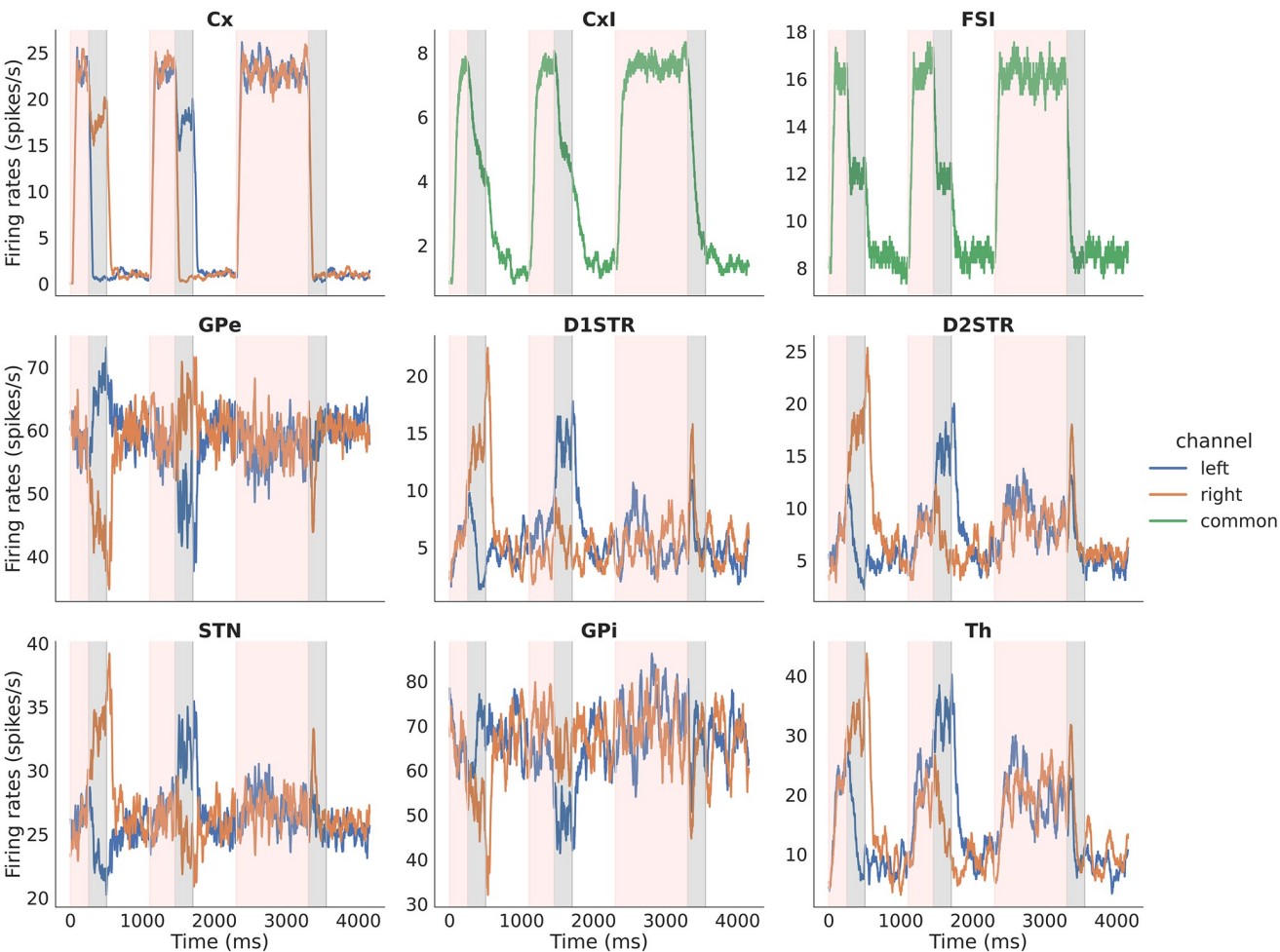

**Fig 3. Example figure showing firing rates for all the nuclei in the n-choice task.** Example figure showing firing rates for all nuclei for three consecutive trials of a 2-choice task, color-coded to distinguish times associated with decision making (pink, *decision phase*) and subsequent times of sustained activity in the selected channel (grey, *consolidation phase*) of each trial. The unshaded regions between each pair of trials are the (*inter-trial interval*). In each panel, the blue (orange) trace corresponds to activity in the left (right) action channel. In this example, the model chooses right on the first trial and left on the second; in the third trial, where decision making times out, no sustained activation is applied to the cortical channel (top left subplot) during the *consolidation phase* (grey region).

In addition to the `firing_rates` data frame, the plotting function `plot_fr` requires the `datatables` (also extracted along with `firing_rates` data frame), the original `results` variable, `experiment_choice` and `display_stim`. The `experiment_choice` ensures that relevant nuclei are plotted and the `display_stim` is a boolean variable that can be set to True/False. When set to True, the stimulation information (e.g., optogenetic or stop signal) is indicated over all trials during which the stimulation was applied. Note that, for a longer simulation, this may slow the plotting function, because the function checks if a stimulation is applied on every trial before indicating the result in the figure. The stop signal application is indicated as a bright horizontal red bar above firing traces of the stimulated nuclei (e.g., Fig 7). The optogenetic stimulation is indicated as a blue bar for excitatory stimulation and yellow for inhibitory stimulation (e.g., Fig 8).

The reward and Q-values data frame can be plotted with the function `plot_reward_Q_df` as follows; note that a figure is shown in the example notebook:

```
reward_fig_handles = plt_func.plot_reward_Q_df(reward_q_df)
```

**2.2.9 Shut down Ray server.** In case the Ray server was selected as the multiprocessing library to use to run the simulation, when the user is done working with the network, the Ray server can be shut down via the terminal with the command

```
ray stop
```

or, if the processes have not all been deactivated, with the command

```
ray stop --force
```

## 2.3 User level modifications

CBGTPy allows for modifications of several parameters that a user can easily perform. All the parameters in the `configuration` variable can be modified. A modified value is usually in the form of a data frame or a dictionary. The underlying function (`ModifyViaSelector`) in `frontendhelpers.py` iterates through all the features listed in the data frame/dictionary and updates the default values with the new values passed to the `configuration` variable. If the user wants to use the default value of a parameter, it is essential to specify the parameter value as `None`, as also shown in Section 2.2. We can subdivide the parameters that can be modified into two major classes, which we will discuss separately: a) parameters related to the agent, which we call the agent parameters (Section 2.3.1); b) parameters related to the experimental environment, which we call the environmental parameters (Section 2.3.2).

Through the adjustment of the appropriate parameters, the user can adapt the model to study a variety of important scientific questions. For example, one could introduce new neural pathways and vary their connectivity to study the effects on the system's dynamics and behavior (e.g., addition of a cortico-pallidal pathway, which may complement stopping mechanisms in the BG pathways [24]). Alternatively, the user could study the effects of specific neural parameters on decision-making and stop signal tasks. Additionally, by introducing optogenetic stimulation to different populations, such as dSPN and iSPN striatal populations, the user could model how the timing and intensity of this stimulation influences the plasticity and learning processes. For example, it has been shown that inhibition of dSPNs during learning impairs performance in a goal-directed learning task [25]. CBGTPy allows stimulation of multiple populations at different phases of a task, which enhances the options for exploring

possible functional pathways and their roles in task performance. Lastly, CBGTPy allows many environmental parameters to be modified for an n-choice or stop-signal task while simulating the activity of the CBGT network. For example, one could test the idea that the slowing down of decision times in healthy humans [26] when there is a high conflict or similarity between choices is related to increased activity in the STN [27]. Taken together, the large number of both network and environmental parameters available for the end user to control is a strength of the CBGTPy framework and greatly increases its ultimate scientific utility.

A detailed list of features of the CBGTPy package that can be easily modified by the user can be found in table S1 Table.

**2.3.1 Agent parameters.** *General neuron parameters.* The parameters common to all neurons can be modified using the `params` field in the dictionary `configuration`. A complete list of editable neuronal parameters is listed in S2 Table. For example, the following dictionary entry can be modified to change the capacitance (`C`) of all neurons:

```
"params": pd.DataFrame([[30]], columns=["C"]),
```

*Population-specific neuronal parameters.* The neuronal parameters of a specific population can be modified using the field `pops` in the dictionary `configuration`. These parameters will override the default values set by the `params` field. A complete list of editable parameters is given in S3 Table. In the following example, the membrane time constant (`Taum`) of the neurons in the FSI population is specified:

```
"pops": {"FSI": {"Taum": [60]}},
```

*Synaptic parameters.* The parameters of the synapses (GABA, AMPA or NMDA) can be modified through the field `receps` in the `configuration` variable. A complete list of editable synaptic parameters is given in S4 Table. In the following example, the membrane time constants of AMPA and GABA synapses (`Tau_AMPA`, `Tau_GABA`) are specified:

```
"receps": pd.DataFrame([[100, 100]], columns=["Tau_AMPA",
"Tau_GABA"]),
```

*Population-specific baseline input parameters.* Each neuron receives a background input from a random Gaussian process with a specified mean frequency and variance equal to 1. Depending on the nature of the background input, the Gaussian process can be excitatory (AMPA and NMDA) or inhibitory (GABA). The user can specify the mean frequency, the efficacy, and the number of connections from the background Gaussian process to the neurons in the population with the dictionary `base`. A complete list of editable input parameters is given

in S5 Table. In the following example, the frequency of the external AMPA inputs (`FreqExt_AMPA`) applied to FSI neurons is specified:

```
"base": {"FSI": {"FreqExt_AMPA": [100]}},
```

*Dopamine-specific parameters.* The dopamine-related parameters can be modified via the `dpmns` parameter, which takes as its input a data frame containing the field name and the names of the parameters to be updated. A complete list of these editable parameters is given in S6 Table. In the following example, the dopamine decay rate (`dpmn_tauDOP`) is specified:

```
"dpmns": pd.DataFrame([[5]], columns=["dpmn_tauDOP"]),
```

*SPN-specific dopaminergic parameters.* The SPN-specific dopaminergic parameters for corticostriatal projections to dSPNs and iSPNs can be modified via `d1` and `d2`, respectively, which take as their input a data frame containing the field name and the parameters to be updated. The complete list of these editable parameters is given in S7 Table. In the following example, the learning rate (`dpmn_alphaw`) and the maximal value for the corticostriatal weights (`dpmn_wmax`) are specified:

```
"dSPN_params": pd.DataFrame([[39.5, 0.08]], columns=
["dpmn_alphaw",
  "dpmn_wmax"]),
"iSPN_params": pd.DataFrame([[-38.2, 0.06]], columns=
["dpmn_alphaw",
  "dpmn_wmax"]),
```

*New pathways.* The parameters of a specific pathway can be changed by using the variable `newpathways`. This variable can also be used to add new connections. This takes as its input a data frame that lists the following features of the pathway: source population (`src`), destination population (`dest`), receptor type (`receptor`), channel-specific or common (`type`), connection probability (`con`), synaptic efficacy (`eff`) and the type of the connection (`plastic`), which can be plastic (`True`) or static (`False`). An example of a cortico-pallidal pathway involving AMPA synapses with 50% connection probability, synaptic strength 0.01, and no plasticity is presented in the following dictionary entry:

```
"newpathways": pd.DataFrame([["Cx", "GPe", "AMPA", "syn", 0.5,
0.01, False]])
```

If the user wishes to change multiple pathways at once, then the variable can be given a list of data frames as input.

*Q-learning process*. CBGTPy uses Q-learning to track the internal representations of the values of the possible choices, which depend on the rewards received from the environment. The parameters of this process can be modified via the variable `Q_support_params`. The two parameters that can be modified are `C_scale` and `q_alpha`. The former controls the scaling between the change in phasic dopamine and the change in weights, and the latter controls the change in choice-specific q-values with reward feedback from the environment. An example of how to specify these two parameters is presented in the following dictionary entry:

```
"Q_support_params": pd.DataFrame([[30, 0.1]], columns=
["C_scale", "q_alpha"]),
```

The equations showing the roles of these parameters are described in detail in S2 Appendix.

*Q-values data frame*. The choices available to the agent can be initialized with identical values (e.g., 0.5) representing an unbiased initial condition. Alternatively, non-default values for the Q-values data frame (such as values biased towards one choice) can be initialized using the variable `Q_df`. An example of how to specify this variable is presented in the following dictionary entry:

```
"Q_df_set": pd.DataFrame([[0.3, 0.7]], columns=["left",
"right"]),
```

Note that this parameter should be updated according to the number of choices specified in the variable `number_of_choices`. The above example shows an initialization for a 2-choice task.

*Cortical activity*. In addition to the background inputs that generate the baseline activity of all of the CBGT nuclei, the cortical component provides a ramping input to the striatal and thalamic populations, representing the presence of some stimulus or internal process that drives the consideration of possible choices. The maximum level of this input can be defined by the parameter `maxstim` and specified using the following dictionary line:

```
"maxstim": 0.8,
```

*Corticostriatal plasticity*. The corticostriatal plasticity in the n-choice experiment can be switched on and off using the `corticostriatal_plasticity_present` parameter. When it is set to `True`, the corticostriatal weights change based on rewards and dopaminergic signals (for more details see S2 Appendix). When it is set to `False`, the simulation proceeds without any update in the corticostriatal weights and the Q-values. The value of this parameter is set to `True` using the following dictionary line:

```
"corticostriatal_plasticity_present": True,
```

*Sustained activation to the action channel for the selected choice*. In order to resolve the temporal credit assignment problem [14], we rely on post-decision sustained activation to keep the selected channel active during the phasic dopaminergic activity [28, 29]. After the choice has been made, following the onset of a trial (*decision phase*), the cortical component of the action channel associated with the selected choice continues to receive inputs, while the unselected channels do not (*consolidation phase*); see Fig 3. This phase may also represent the movement time of the agent. The assumption here is that this activation provides an opportunity for corticostriatal plasticity that strengthens the selected choice. The parameter `sustainedfraction` is the fraction of input stimulus maintained during the consolidation phase in the cortical channel corresponding to the action selected by the agent, and it can be specified using the following dictionary entry:

```
"sustainedfraction": 0.7,
```

Once the dopamine signal has been delivered at the end of the *consolidation phase*, all cortical inputs are turned off for the *inter-trial interval*. See Section 3.1 and Fig 4 for more details on the trial phases.

*Thalamic threshold*. In the default set-up, when the thalamic firing rate of either choice reaches 30 Hz, that choice is selected. This threshold can be specified by the user by setting the parameter `thalamic_threshold` in the following way:

```
"thalamic_threshold": 30,
```

**2.3.2 Environment parameters.** *Experiment choice* The parameter `experiment_choice` is set at the beginning of the simulation (see Section 2.2). It also needs to be sent as a configuration variable, so that the specific functions and network components relevant to the appropriate experiment are imported.

*Inter-trial interval*. The parameter `inter_trial_interval` allows the user to specify the inter-trial interval duration. The inter-trial interval also corresponds to the duration of the *inter-trial-interval phase* of the simulation, where the network receives no external input and

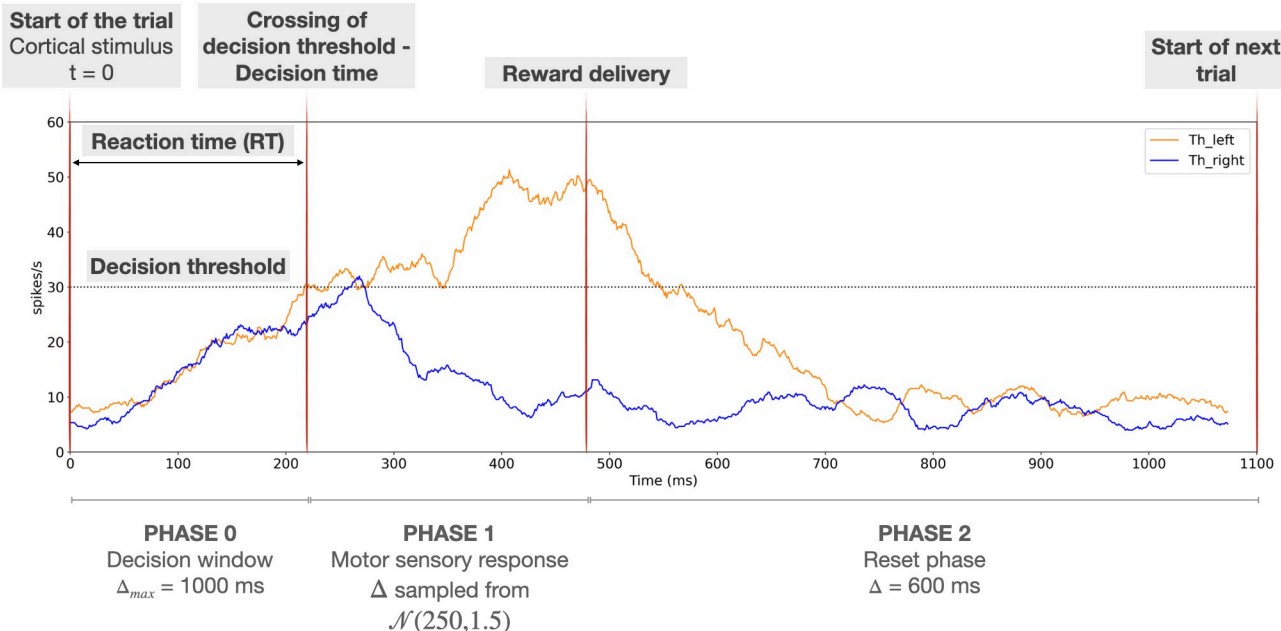

**Fig 4. Representation of the different phases of the simulated decision process.** This sketch represents the simulation of one trial of the 2-choice task in which a left choice is made. The first red vertical line, at time 0, represents the time of onset of a ramping stimulus to $Cx_i$ for both $i$, which indicates the start of the trial. The second red vertical thick line depicts the decision time (end of *phase 0*). The third red vertical line depicts the end of the motor response period associated with the decision (end of *phase 1*), which is also the time when reward delivery occurs and hence dopamime level is updated. After this time, the reset phase (*phase 2*) starts; this ends after 600 *ms* (inter trial interval), when a new trial starts (right-hand vertical red line). Orange and blue traces represent the mean thalamic firing rates $Th_i$ for $i \in$ {left, right}, respectively, and the horizontal black dotted line highlights the decision threshold.

shows spontaneous activity. When no value is specified (None), a default value of 600 *ms* is used. The user can set the value of this parameter using the following dictionary entry:

```
"inter_trial_interval": 600,
```

*Movement time*. After a choice is made, the chosen action channel receives sustained activation at some fraction (with a default value of 70%) of the initial cortical input strength. As noted in the previous section, this phase of the simulation (*consolidation phase*) represents the movement time, which is distinct from the reaction time, provides a key window for corticostriatal synaptic plasticity to occur, and remains unaffected by the selected choice. The length of this phase can be controlled with the parameter movement_time. The default value of the movement time (when this parameter is set to None) is sampled from a normal distribution $\mathcal{N}(250, 1.5)$. However, the user can choose to set it to a constant value by passing a list ["constant", N], where N represents the constant value of movement time for all trials. The other option is to sample from a normal distribution of specified mean N using ["mean", N]. The movement time can be set to a fixed value as follows:

```
"movement_time": ["constant", 300],
```

*Choice time out.* The parameter `choice_timeout` controls the duration of the time interval in which a choice can be made. The default value of this parameter (when it is set to `None`) is 1000 *ms*. This parameter can be changed as follows:

```
"choice_timeout": 300,
```

*Choice labels.* The data frame `channels` allows the labels for the action channels to be changed. The new labels can be used to access information about the action channels. An example is shown below:

```
"channels": pd.DataFrame([["left"], ["right"]], columns=
["action"]),
```

Note that this parameter should be updated according to the number of choices specified in the variable `number_of_choices`. The above example shows an initialization for a 2-choice task.

*Number of trials.* The `n_trials` parameter sets the number of trials to be run within a simulation. Note that this number should be greater than the `volatility` parameter (described in the following paragraph). However, if only 1 trial is to be simulated, then `volatility` parameter should be set to `None`. Examples of how to set this parameter are as follows:

```
"n_trials": 2,
```

```
"n_trials": 1,
...
"volatility": [None, "exact"]
```

For more details about setting `volatility` parameter, please refer to the following paragraph.

*Volatility*. The parameter `volatility` indicates the average number of trials after which the reward contingencies switch between the two choices. The volatility parameter is a list consisting of two values, [λ, 'option'], where `option` can be set as `exact` or `poisson`. The λ parameter generates a reward data frame where the reward contingency changes after an average of λ trials. The option `exact` ensures that the reward contingency changes exactly after λ trials whereas the option `poisson` samples the change points from a Poisson distribution with parameter λ. However, note that this parameter cannot be 0 or the total number of trials. To perform a simulation in which the reward contingencies do not change until the end of the simulation, set this parameter to `n_trials−1` and drop the last trial from the analysis. An example of how to define and specify the volatility is shown in the following command line:

```
"volatility": [2, "exact"],
```

Note that for a 1-choice task or stop signal task, the volatility parameter is not applicable and hence should be defined differently; specifically, the parameter λ should be set to None as follows:

```
"volatility": [None, "exact"],
```

*Reward probability*. The parameter `conflict` represents the reward probability of the reward data frame and is defined as a tuple of reward probabilities for the *n* choices. In the following example, for a 2-choice task, the first reward probability corresponds to the first choice listed in the `channels` parameter (e.g., "left"). The reward probabilities for the choices are independent, thereby allowing reward structure to be set in the format (p1, p2) as in the two following examples, representing unequal and equal reward probabilities, respectively:

```
"conflict": (0.75, 0.25),
"conflict": (0.75, 0.75),
```

Note that this parameter should be updated according to the number of choices specified in variable `number_of_choices`. The above example shows an initialization for a 2-choice task. For example, for a 3-choice task the reward probabilities can be defined as:

```
"conflict": (1.0, 0.5, 0.2),
```

*Reward parameters.* The trial-by-trial reward size is generated by a random Gaussian process, with a mean (`reward_mu`) and a standard deviation (`reward_std`) that can be assigned. To simulate binary rewards, choose mean = 1 and standard deviation = 0, as follows:

```
"reward_mu": 1,

"reward_std": 0.0,
```

*Optogenetic signal.* If the experiment requires an optogenetic signal to be applied, then this should be indicated in the `configuration` variable by setting `opt_signal_present` to `True`, with each boolean variable corresponding to each nucleus specified in the list of populations to be stimulated, as shown below:

```
"opt_signal_present": [True],

...

..

"opt_signal_population": ["dSPN"],
```

The above example shows the case, when only one population (i.e dSPNs) is stimulated. More than one population can be stimulated simultaneously as shown below:

```
"opt_signal_present": [True, True],

...

..

"opt_signal_population": ["dSPN", "iSPN"],
```

*Optogenetic signal probability.* The `opt_signal_probability` parameter accepts either a float or a list. The float represents the probability of the optogenetic signal being applied in any given trial. For example, the user wants all the trials in the simulation to be optogenetically stimulated, i.e `opt_signal_probability` = 1.0, it should be defined as shown below:

```
"opt_signal_probability": [1.0],
```

Alternatively, a specific list of trial numbers during which optogenetic stimulation should be applied can also be passed. An example is shown below:

```
"opt_signal_probability": [[0, 1]],
```

In the above example, a list of trial numbers [0, 1] indicates that the optogentic stimulation is applied to trial numbers 0 and 1.

Please note that if more than one nucleus will be stimulated, the opt_signal_probability expects a list of floats (probabilities) or list of list (list of trial numbers). For example, as mentioned in an example above, say the user wants to stimulate two populations (dSPN and iSPN) at trial numbers [0, 1] and [1, 2] respectively:

```
"opt_signal_present": [True, True],
..
"opt_signal_population": ["dSPN", "iSPN"],
"opt_signal_probability": [[0, 1],[1, 2]],
```

*Optogenetic signal amplitude*. The amplitude of the optogenetic signal can be passed as a list of floats to the parameter opt_signal_amplitude. A positive value represents an excitatory optogenetic signal, whereas a negative value represents an inhibitory optogenetic signal. An example of excitation is shown below:

```
"opt_signal_amplitude": [0.1],
```

If we want to send different amplitudes of optogenetic signals to different populations, for example, an excitatory (0.3) to dSPNs and inhibitory (−0.25) to iSPNs, then the amplitudes should be specified as:

```
"opt_signal_present": [True, True],
..
"opt_signal_population": ["dSPN", "iSPN"],
"opt_signal_amplitude": [0.3, -0.25],
```

*Optogenetic signal onset*. The `opt_signal_onset` parameter sets the onset time for the optogenetic signal. The onset time is measured relative to the start of the *decision phase*. For example, to specify that optogenetic stimulation will start 20 *ms* after the *decision phase* begins, the appropriate command is:

```
"opt_signal_onset": [20.],
```

*Optogenetic signal duration*. The duration for which the optogenetic signal is applied can be controlled by the parameter `opt_signal_duration`. This parameter accepts a numerical value in *ms* as well as phase names as strings. For example, to apply the optogenetic stimulation for 1000 *ms* after the signal onset, the command is:

```
"opt_signal_duration": [1000.],
```

However, in order to apply optogenetic stimulation during the whole *decision phase*, the duration variable should be the string "phase 0" as shown below:

```
"opt_signal_duration": ["phase 0"],
```

This allows the user to specifically target *decision* ("phase 0"), *consolidation* ("phase 1") and *inter-trial interval* ("phase 2") phases with optogenetic stimulation.

*Optogenetic signal channel*. The user can also control whether the optogenetic signal is applied globally to all action channels (`all`), to a randomly selected action channel (`any`), or to a specific action channel (for instance, `left`). To do so, the parameter `opt_signal_channel` needs to be specified as in the example below:

```
"opt_signal_channel": ["all"],
```

*Optogenetic signal population*. The optogenetic stimulation can be applied to a single or multiple populations in the same simulation. In either case, the population names should be defined as a list. The target population can be specified using the parameter `opt_signal_population`. In the example below, the dSPN population is set as the target population:

```
"opt_signal_population": ["dSPN"],
```

Although the optogenetic-related parameters can be used to mimic the effect of a stop signal manifested as the application of a step input current to a population, the network also has a number of modifiable parameters that are specific to injecting the stop signal to target nuclei via a box-shaped current.

*Recorded variables*. CBGTPy allows recording of time-dependent values of both the corticostriatal weights and any optogenetic inputs to any CBGT nuclei that are being stimulated by using the parameter `record_variables`. The first component of `record_variables` can be used to track the evolution of the weights from cortex to dSPNs or iSPNs during an n-choice experiment. The latter component records the optogenetic input applied to the target population and is especially useful for debugging purposes. Both of these variables can be extracted as a data frame by calling the function `extract_recording_variables` as described in Section 2.2. Note that, for the parameter `weight`, cortical weights to both dSPNs and iSPNs for all choice representations (channels) are recorded. In addition, when running the stop signal task, the stop signal inputs can be recorded as well. Here, we can see an example of how to extract the weights and the optogenetic input:

```
"record_variables": ["weight", "optogenetic_input"],
```

In addition, for the stop signal task, CBGTPy also allows the recording of the variable "`stop_input`", which can be used to check if the stop signal inputs were applied correctly to the target nuclei.

```
"record_variables": ["stop_input"],
```

*Stop signal*. If the experiment requires the stop signal to be applied, then this option should be selected in the `configuration` variable by setting `stop_signal_present` to `True`. Different stop signals can be applied to different target populations during the same execution. For these, the `stop_signal_present` variable is defined as a list whose length depends on the number of target populations. For example, if we apply stop signals to two different populations, we have to set this variable as follows:

```
"stop_signal_present": [True, True],
```

Note that the user can apply the stop signals to as many nuclei as desired.

*Stop signal populations.* The target populations for the stop signal can be specified using the parameter `stop_signal_population`. In the example below, the STN and GPeA populations are set as the target populations:

```
"stop_signal_populations": ["STN", "GPeA"],
```

All the examples presented below corresponding to the stop signal task are designed taking into account that the stop signal is injected into these two populations.

*Stop signal probability.* The stop signal probability can be specified using the parameter `stop_signal_probability`, which is a list whose entries can take a float or a list as input. If a float (between 0 and 1) is introduced, then this value represents the probability to which the stop signal is applied. These trials are picked randomly from the total number of trials. Alternatively, if a sublist is specified, then it must contain the numbers of those trials where the user wants the stop signal to be applied. Note that the entries in the main list refer to the populations specified in the same order as in the variable `stop_signal_populations`. For example, within the statement

```
"stop_signal_probability": [1.0, [2, 3, 6]],
```

the first float value represents a 100% probability of applying the "first" stop signal to the first specified nucleus (STN), while the subsequent list of values represents the numbers of the trials on which the "second" stop signal will be applied to the other target region (GPeA).

*Stop signal amplitude.* The amplitude of the stop signal can be passed as a float to the parameter `stop_signal_amplitude`. Note that this parameter is a list and that every value refers to the corresponding population. The order should follow the order of the populations. For example,

```
"stop_signal_amplitude": [0.4, 0.6],
```

*Stop signal onset.* The parameter `stop_signal_onset` sets the times when the stop signals are injected into the target nuclei. The onset time is measured with respect to the start of the *decision* phase. In the example proposed below, the stop signal stimulation at the STN starts 30 *ms* after the *decision* phase begins, while that applied to the GPeA starts 60 *ms* after the *decision* phase begins:

```
"stop_signal_onset": [30., 60.]
```

*Stop signal duration.* How long each stop signal is maintained can be controlled using the parameters `stop_signal_duration`. As in the previous cases, this is a list whose order must follow that of the target populations. In the following example, a stop signal lasting 100 *ms* is applied to the STN while another with duration 160 *ms* is applied to the GPeA:

```
"stop_signal_duration": [100.,160.]
```

In order to apply the stop signal throughout an entire phase, the duration variable should be set to a string containing the name of the phase when the user wants to apply the stop signal, as shown below:

```
"opt_signal_duration": ["phase 0", "phase 1"],
```

This allows the user to specifically target *decision* ("phase 0"), *consolidation* ("phase 1") and *inter-trial interval* ("phase 2") phases with persistent stop signal stimulation.

*Stop signal channel.* The user can also control if the stop signal applied to a specific population is presented to all action channels (`all`), to a uniformly and randomly picked action channel (`any`), or to a specific action representation (for instance, `left`). These can be specified using the parameter `stop_signal_channel`. In the following example, one stop signal is presented to the STN populations of all of the action channels, while the second stop signal is applied only to the GPeA population corresponding to the "left" action channel:

```
"stop_signal_channel": ["all", "left"]
```

## 3 Experiments

In this section, we present some details about examples of the two primary experiments that CBGTPy is designed to implement.

### 3.1 An n-choice task in an uncertain environment

This task requires the agent to select between *n* choices (e.g., left/right in a 2-choice task). The selection of each choice leads to a reward with a certain probability. Moreover, the reward probability associated with each choice can be abruptly changed as part of the experiment. Thus, there are two forms of environmental uncertainty associated with this task: a) *conflict*, or the degree of similarity between reward probabilities; and b) *volatility*, or the frequency of changes in reward contingencies. Higher conflict would represent a situation where the reward

probabilities are more similar across choices, making detection of the optimal choice difficult, whereas a lower conflict represents highly disparate values of reward probabilities and easier detection of the optimal choice. Conflict is not specified directly in CBGTPy; rather, the reward probabilities are explicitly set by the user (Section 2.3.2). An environment with high (low) volatility corresponds to frequent (rare) switches in the reward contingencies. The volatility can be set by the parameter λ, which determines the number of trials before reward probabilities switch. The user can choose between whether the trials are switched after exactly λ trials or whether switches are determined probabilistically, in which case λ represents the rate parameter of a Poisson distribution that determines the number of trials before reward probabilities switch (Section 2.3.2).

Using the reward probabilities and volatility, the backend code generates a reward data frame that the agent encounters during the learning simulation. The reward data frame is used in calculating the reward prediction error and the corresponding dopaminergic signals, which modulate the plasticity of the corticostriatal projections.

At the beginning of the simulation, the CGBT network is in a resting phase during which all CBGT nuclei produce their baseline firing rates. When a stimulus is presented, the network enters a new phase, which we call *phase 0* or *decision* phase. We assume that at the start of this stage a stimulus (i.e., an external stimulus, an internal process, or a combination of the two) is introduced that drives the cortical activity above baseline. Cortical projections to the striatal populations initiate ramping dynamics there, which in turn impacts activity downstream in the rest of the BG and Th. We also assume that when the mean firing rate of a thalamic population exceeds a designated threshold value of 30 spikes per second (the so-called decision boundary), the CBGT network, and hence the agent, has made a choice. This event designates the end of *phase 0*, and the duration of this phase is what we call the *reaction time*. If a decision is not made within a time window $\Delta_{max}$ ms after the start of the phase, then we say that none of the available choices have been selected and the decision is recorded as "none". Such trials can be excluded from further analysis depending on the hypothesis being investigated.

To allow for selection between *n* different choices we instantiate *n* copies of all CBGT populations except *FSI* and *CxI*. This replication sets up action channels representing the available choices that can influence each other indirectly through the shared populations and otherwise remain separate over the whole CBGT loop. To distinguish between the firing ratse of the populations within channels, we will call them $Pop_i$, where *Pop* refers to the corresponding CBGT region and *i* refers to the channel name (e.g., $Cx_{left}$, $Cx_{right}$ for $n = 2$).

The presentation of a stimulus to the cortical population is simulated by increasing the external input frequency in all copies of the cortical *Cx* populations that ramp to a target firing rate $I_{target}$. The ramping current $I_{ramp}(t)$ is calculated as

$$I_{ramp}(t) = I_{ramp}(t - dt) + 0.1[I_{target}(t) - I_{ramp}(t - dt)]$$

where *dt* is the integrator time step, and the external input frequency also changes according to

$$f_{ext,x}(t) = f_{ext,x,baseline}(t - dt) + I_{ramp}(t).$$

After a *Th* population reaches the threshold and hence a decision is made, the ramping input to *Cx* is extinguished and a subsequent period that we call *phase 1* or *consolidation* phase begins, which by default has duration sampled from a normal distribution $\mathcal{N}(\mu = 250 \ ms, \sigma = 1.5 \ ms)$ but can also be fixed to a given duration. This phase represents the period of the motor implementation of the decision. Activity during *phase 1* strongly depends on what happened in *phase 0* such that, if a decision *i* occurred at the end of *phase 0*, then $Cx_i$ will be induced to

exhibit sustained activity during *phase 1* [28] (see S2 Appendix), while in any non-selected action channels, the cortical activity returns to baseline. If no decision has been made by the network (within a time window $\Delta_{max}$ *ms*, with a default value of 1000 *ms*), then no sustained activity is introduced in the cortex (see Fig 3, 3rd trial, the top left subplot showing cortical activity).

Finally, each trial ends with a reset phase of duration 600 *ms* (although this can be adjusted by the user), which we call *phase 2* or the *inter-trial interval* phase, when the external input is removed and the network model is allowed to return to its baseline activity, akin to an inter-trial interval.

A visualization of the decision phases is shown in Fig 4, where two different options, *right* and *left*, are considered. The blue trace represents the thalamic activity for the *right* channel, $Th_{right}$, while the orange trace represents that for the *left* channel, $Th_{left}$. At the end of *phase 0*, we can see that $Th_{left}$ reaches the decision threshold of 30 spikes per second before $Th_{right}$ has done so, resulting in a left choice being made. During *phase 1*, the $Th_{left}$ activity is maintained around 30 spikes per second by the sustained activity in $Cx_{left}$.

In this task, critical attention should be paid to *phase 0*, as this represents the process of evidence accumulation where the cortical input and striatal activity of both channels ramp until one of the thalamic populations' firing rates reaches the threshold of 30 *Hz*. To be largely consistent with commonly used experimental paradigms, the maximal duration of this phase is considered to be $\Delta_{max} = 1000$ *ms* such that, if the agent makes no decision within 1000 ms, the trial times out and the decision is marked as "none". These trials can be conveniently removed from the recorded data before analysis. If a decision is made, then the simulation proceeds as though a reward is delivered at the end of *phase 1*—that is, at the end of the motor sensory response—such that *phase 1* represents the plasticity phase, where the choice selected in *phase 0* is reinforced with a dopaminergic signal. During this phase, the cortical population of the selected channel receives 70% of the maximum cortical stimulus applied during the ramping phase, although the user can change this percentage. This induces sustained activity that promotes dopamine- and activity-dependent plasticity as described in S2 Appendix. The activity-dependent plasticity rule strengthens (weakens) the corticostriatal weight to dSPNs (iSPNs) of the selected channel when dopamine rises above its baseline level. CBGTPy allows for the specification of other parameters such as learning rate, maximum weight values for the corticostriatal projections and dopamine-related parameters (see details with examples in Section 2.3).

At the beginning of the simulation, with baseline network parameters, the selection probabilities are at chance level (i.e., 50% for a 2-choice task). If the network experiences rewards, however, the dopamine-dependent plasticity strengthens the corticostriatal projection to the dSPN population of each rewarded choice, thereby increasing the likelihood that it will be selected in the future. CBGTPy allows for probabilistic reward delivery associated with each option, as well as switching of these probabilities between the two actions (Fig 5). When such a change point occurs, the previously learned action now elicits a negative reward prediction error, forcing the network to unlearn the previously learned choice and learn the new reward contingency.

## 3.2 A stop signal task

The stop signal task represents a common paradigm used in cognitive psychology and cognitive neuroscience for the study of reactive inhibition [30]. In this task, participants are trained to respond as fast as possible after the presentation of a "Go" cue. Sometimes the "Go" cue is followed by the presentation of a "Stop" cue, which instructs subjects to withhold their decision and hence, if successful, prevents any corresponding movement before it begins.

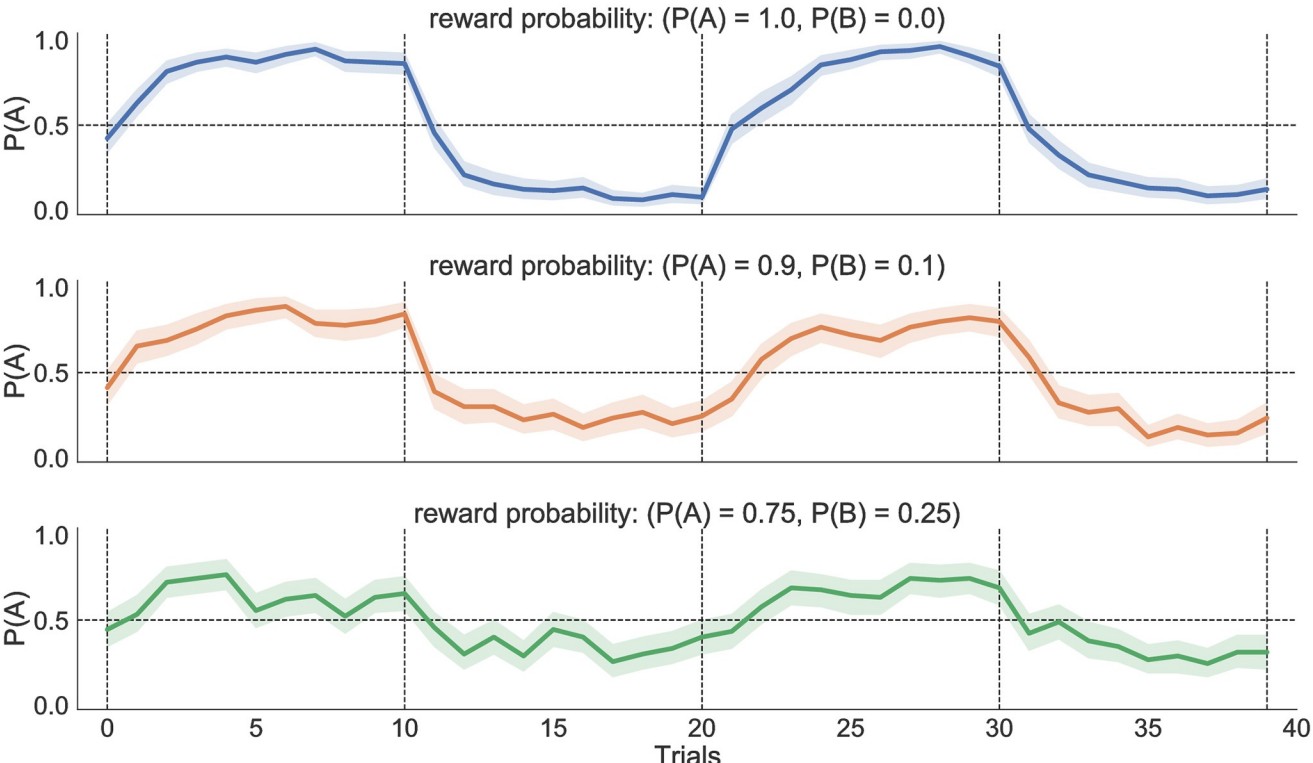

**Fig 5. Probability of choosing the more rewarded action (e.g., A or B) for different levels of conflict in a 2-choice task.** The reward contingencies flip between the two choices every 10 trials (marked by vertical dashed lines), at which point the probability of choosing the more rewarded option drops below chance. The probability of choosing *A* is high in the 1st and 3rd blocks; however, the probability to choose *A* drops in the 2nd and 4th blocks (where *B* is rewarded with a higher probability). Performance, measured in terms of probability of selecting option *A*, degrades in general as conflict increases, but sensitivity to change points drops. The performance was averaged over 50 random seeds for each conflict level.

Imaging and electrophysiological studies in humans, rodents, and monkeys agree in reporting that STN neurons become activated in response to a stop signal [31], providing a fast, non-selective pause mechanism that contributes to action suppression through the activation of the cortical hyperdirect pathway [32, 33]. However, this mechanism, by itself, mostly fails to inhibit locomotion, appearing to be not selective and long-lasting enough to prevent a late resurgence of the evidence accumulation process as needed to guarantee a complete cancellation of the execution of the motor response [34]. A complementary slower but selective mechanism is thought to be provided by the activation of arkypallidal neurons in the GPe in response to an external stimulus that instructs the network to brake the ongoing motor planning process [33–35]. According to this idea, a long-lasting action inhibition results from the activation of pallidostriatal GABAergic projections. For more details on how this mechanism takes place see [12].

To reproduce these mechanisms and to simulate the interruption of the action selection process, we inject two independent, external, excitatory currents directly into STN and GPeA neurons during a typical CBGT simulation. This choice is based on the findings of Mallet et. al. (2016) [34]. The stop signal is excitatory and hence is simulated by up regulating the baseline input frequency to the AMPA receptors

$$f_{ext,x}(t) = f_{ext,x}(t) + \texttt{stop\_amplitude},$$

where `stop_amplitude` defines the magnitude of the stop signal stimulation. The currents injected as a step function cause an increase in the firing rates of the target nuclei. Both of these external currents are defined using parameters that can be modified in an easy and user-friendly way, without requiring any familiarity or advanced knowledge of the details of the implementation (see Section 2.3 for further details and examples).

The following list of parameters characterizes each one of these currents:

(a) `amplitude`, specifying the magnitude of the stimulation applied;

(b) `population`, specifying the CBGT region or sub-population being stimulated.

(c) `onset`, specifying the onset time of the stimulation, with respect to the trial onset time (i.e., the beginning of *phase 0*);

(d) `duration`, defining the duration of the stimulation. Since the length of *phase 0* is not fixed and is dependent on how long it takes for the thalamic firing rates to reach the decision threshold (30Hz), this parameter should be set carefully;

(e) `probability`, determining the fraction of trials on which the stop signal should be introduced;

(f) `channel`, defining which action channels are stimulated.

A more detailed description of all of these parameters can be found in Section 2.3.2. The details of the stop parameters used to reproduce Fig 7 are included in S9 Table.

The characterization of the different network phases described in Section 2 slightly changes when performing the stop signal task (see Fig 6). During the *decision* phase (*phase 0*, which in this task lasts for a maximum of 300 ms) the stop signals are directly presented to the target populations by injecting independent external currents. The user can choose the moment of the injection by manipulating the variable `stop_signal_onset_time`. These signals are kept active for a period equal to `stop_signal_duration`, with a magnitude equal to the `stop_signal_amplitude`. These values do not need to be the same for all of the stop signals used. At this stage, two possible outcomes follow: (a) despite the presentation of the stop signals, the network still manages to choose an action; or, (b) the network is not able to make an action after the presentation of the stop signals, and *phase 0* ends with no action triggered, which represents a `stop` outcome. The former option could arise for various reasons; the strength of the stop signal may not be sufficient to prevent the network from triggering an action or the evaluation process may still have enough time to recover, after the stop signal ends, to allow the thalamic firing rates to reach the decision threshold (e.g., 30 *Hz*) within the permitted decision window.

In Fig 7 we show an example of stop signal stimulation applied to STN and GPeA populations, independently, in a 1-choice task. The onset of the stimulation applied to STN occurred at 30 *ms* while that for the stimulation of GPeA was set to 60 *ms*; both signals were applied for a duration of 145 *ms*. Both stimulations were applied in both of the trials shown. Note that trial number 1 corresponds to a correct stop trial (no decision was made within *phase 0*), whereas the following trial corresponds to a failed stop trial. These outcomes can be inferred from the activity of various populations at the end of the decision making windows: the thalamic firing rate reaches a higher level there and the firing traces in GPi decrease more for the failed stop trial than for correct stop, while cortical activity is sustained beyond this time specifically when stop fails.

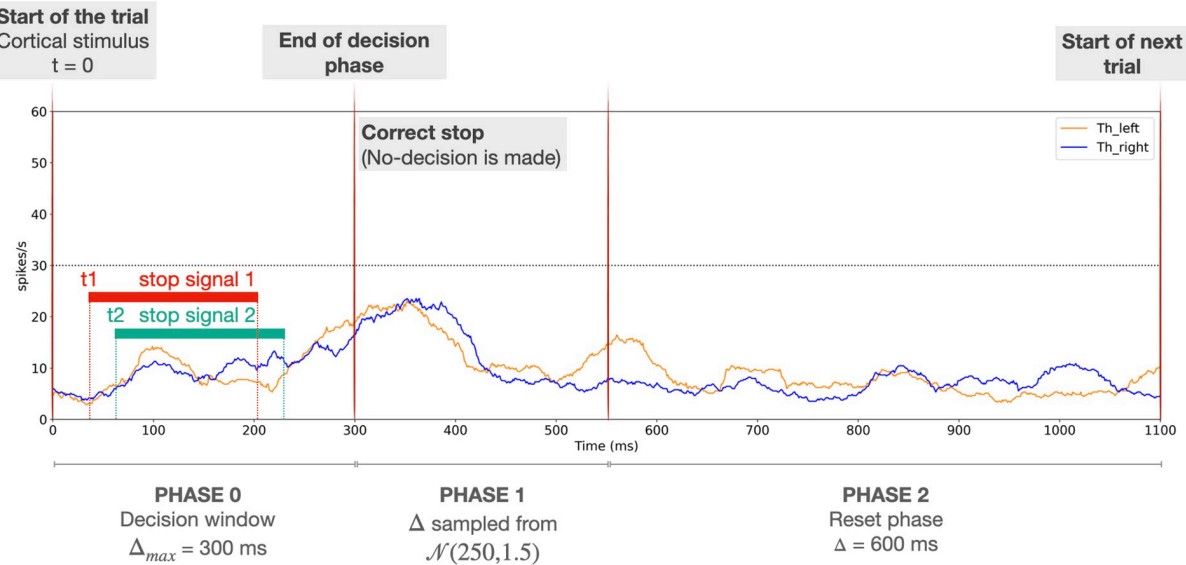

**Fig 6. Representation of the phases of the stop signal task with two action channels.** This sketch represents one trial of the simulation. The first red vertical line indicates the presentation of the cortical stimulus (causing ramping of cortical activity), which represents the start of the trial. Red and green horizontal bars depict the presentation of two stop currents, according to the onset time and duration values chosen by the user. In this example, the two stop signals are considered to be applied with different onset times $t_1$ and $t_2$, respectively. The second red vertical line depicts either the moment when an action has occurred (end of *phase 0*) or that 300 *ms* has expired and no action has been triggered, so a successful stop has occurred. The third red vertical line depicts the end of the motor sensory response phase (end of *phase 1*), if an action is triggered (failed stop). Here, the stop was successful (no decision threshold crossing within the decision window), so no motor sensory response is visible. Finally, the *reset phase* (*phase 2*) occurs, after which a new trial begins. Blue and orange traces represent the mean thalamic firing rates $Th_x$ for $x \in$ {right,left}, respectively, and the horizontal black dotted line marks the decision threshold.

## 3.3 Optogenetic stimulation

CBGTPy also allows for simulation of optogenetic stimulation of CBGT nuclei while an agent performs the available tasks (stop signal or n-choice). Optogenetic stimulation is implemented by setting a conductance value for one of two opsins dependant upon the mode of stimulation, channelrhodopsin-2 for excitation and halorhodopsin for inhibition. The excitatory or inhibitory optogenetic input is applied as a current $I_{opto}$ added to the inward current $I_{ext}$ of all neurons in a nucleus or subpopulation during a typical CBGT simulation such that

$$I_{opto}(t) = \begin{cases} g_{opto}(V(t) - V_{ChR2}) & g_{opto} \geq 0 \\ -g_{opto}(V(t) - V_{NpHR}) & g_{opto} < 0 \end{cases}$$

where the conductance $g_{opto}$ is a signed value entered via the configuration variable in the notebooks. The reversal potential of channelrhodopsin ($V_{ChR2}$) was considered to be 0 $mV$ and halorhodopsin ($V_{NpHR}$) was considered to be −400 $mV$ [36].

The stimulation paradigm includes the following parameters:

(a) `amplitude`, the sign of which specifies the nature (positive→excitatory / negative→inhibitory) and the absolute value of which specifies the magnitude of the conductance applied;

(b) `population`, specifying the CBGT region or sub-population being stimulated;

(c) `onset`, specifying the onset time of the stimulation;

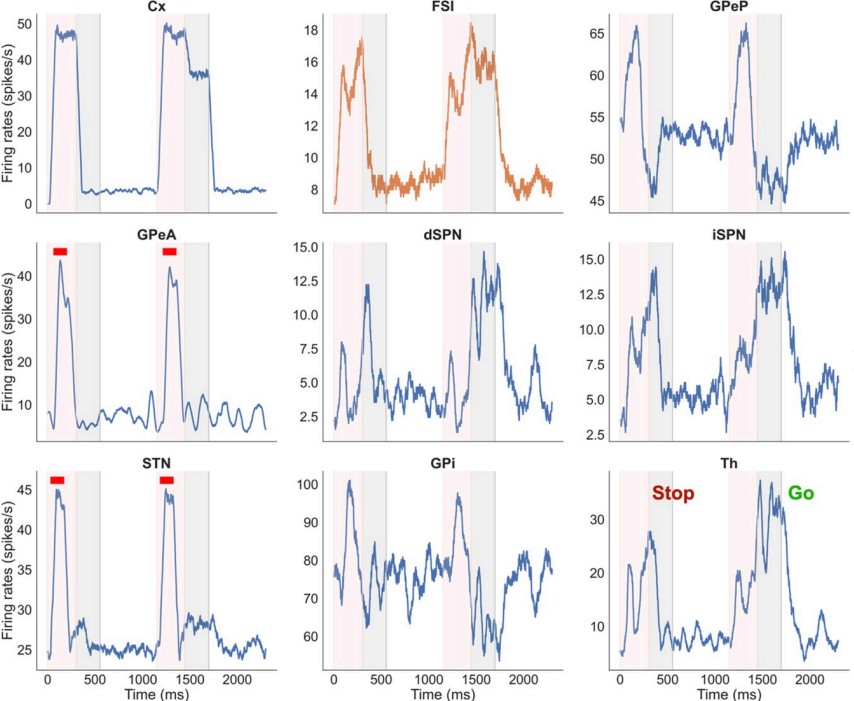

**Fig 7. Example figure showing firing rates for all nuclei for two consecutive stop trials.** Note that the simulation has been run in a 1-channel regime and two stop currents have been applied to STN and GPeA, respectively (see thick red bars). Segments of the simulation are color-coded to distinguish times associated with decision making (pink, phase 0) and subsequent times of motor response (grey, phase 1, showing sustained activity in the selected channel when a decision is made) in each trial. The unshaded regions after the trials are the inter-trial-intervals (phase 2).

(d) `duration`, defining the duration of the stimulation;

(e) `probability`, indicating the fraction of trials or a list of trial numbers to include stimulation;

(f) `channel`, specifying which action channels are stimulated.

The parameter `population` should be entered as a list of the subpopulations to be stimulated. The parameter `onset` is calculated from the beginning of *phase 0*; for example, if this parameter is 10, then the optogenetic stimulation starts 10 *ms* after *phase 0* starts. The parameter `duration` controls the duration of the optogenetic stimulation. This parameter either accepts a numeric value in ms or a string specifying which *phase* should be stimulated. The numeric value stipulates that the list of selected populations will be stimulated from the specified onset time for the specified time duration. The string (e.g., "phase 0") stipulates that the stimulation should be applied throughout the specified phase, thereby allowing the user to specifically target the *decision*, *consolidation* or *inter-trial interval* phase. If an optogenetic configuration results in extending the duration of a phase (e.g., strongly inhibiting dSPN may extend *phase 0*), a time out is specified for every phase to prevent a failure to terminate the phase. The default timeouts for *phase 0, 1* and *2* are 1000 *ms*, 300 *ms* and 600 *ms* respectively unless specified by the user.

The parameter `probability` offers the flexibility of either assigning a number that determines the fraction of trials (randomly sampled from the full collection of trials) on which the stimulation is to be delivered or else entering a list of specific trial numbers. Lastly, the

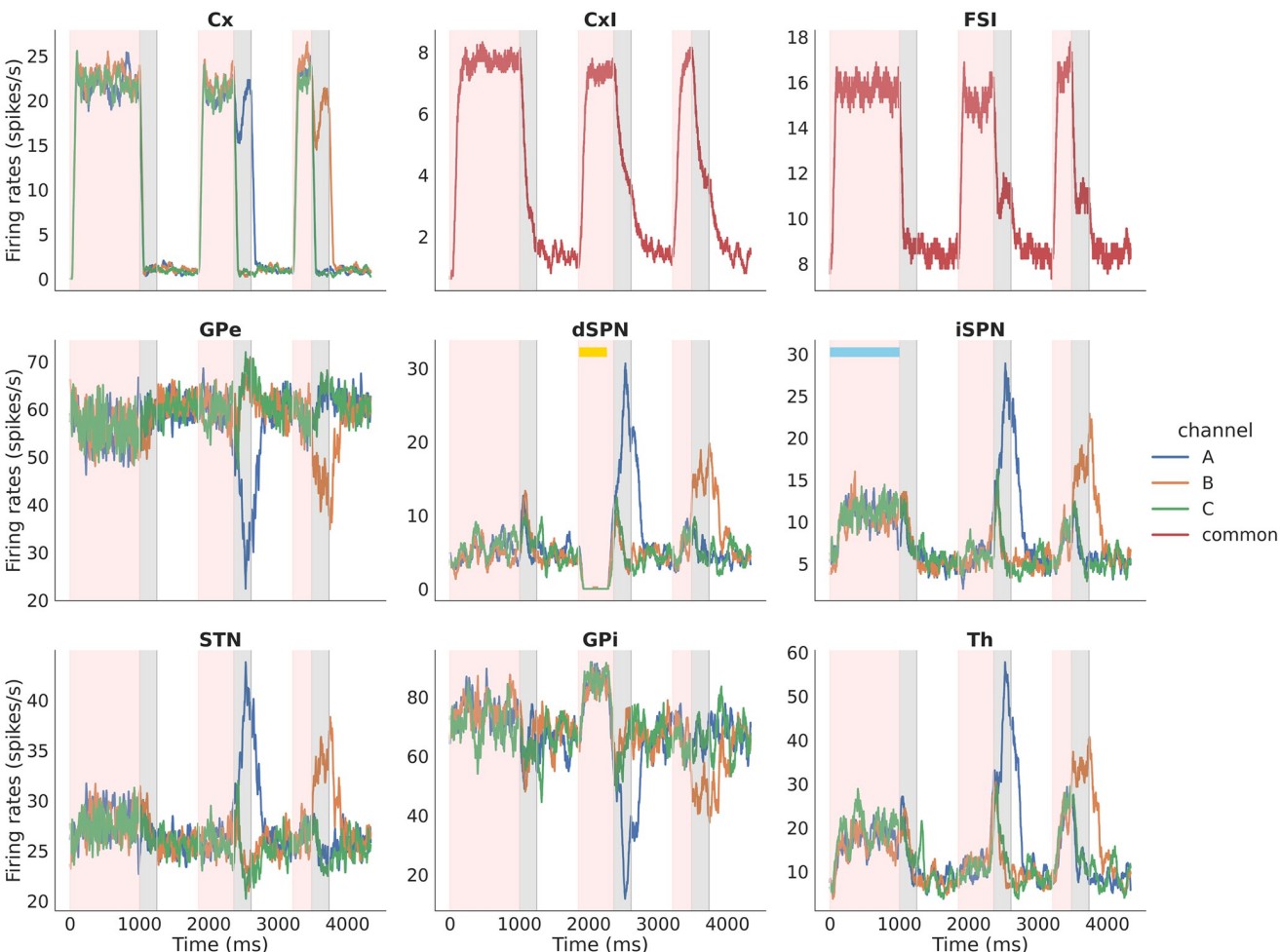

**Fig 8. Example figure showing optogenetic stimulation for the nuclei 'iSPN' and 'dSPN'.** The configuration specified was: `amplitude: [0.5, -0. 5]`, `duration: ['phase 0', 400]`, `trial numbers: [[0], [1]]`, `channels: ['all', 'all']`. The excitatory optogenetic stimulation given to iSPN (shown as blue bar) and lasts all through *phase 0*, whereas inhibitory stimulation to dSPN (shown as yellow bar) lasted for 400 *ms*. In both cases, stimulation were applied to all channels (namely A, B and C) of the nuclei.

parameter `channel` specifies the name of the action channel, such as "left", onto which the stimulation should be applied. This parameter also accepts two additional options, "all" or "any", the former of which leads to the application of a global stimulation to the same population in all channels and the latter of which randomly selects a channel for stimulation on each trial. The details of the optogenetic input is included in S10 Table.

We show an example of optogenetic stimulation applied to a list of iSPN and dSPN populations in a 3-choice task (Fig 8). The excitatory stimulation (shown as thick blue bar), with an amplitude of 0.1, was applied to the iSPN populations of all the channels (namely A, B, and C) in the first trial, for the duration of *phase 0*. An increased activity in the iSPN population (activation of the indirect pathway) caused a choice time out on this trial. In the subsequent second trial, an inhibitory stimulation with an amplitude of −0.5 was applied to the dSPNs (shown as yellow bar) for 400 *ms*. This resulted in brief but strong inhibition of dSPNs (direct pathway), thereby delaying the action selection. This can be observed by comparing the durations of the *decision phase between the second and third trials*, where no such manipulation was imposed.

## 4 Discussion

Here we introduce CBGTPy, an extensible generative modeling package that simulates responses in a variety of experimental testing environments with agent behavior driven by dynamics of the CBGT pathways of the mammalian brain. A primary strength of this package is the separation of the agent and environment components, such that modifications in the environmental paradigm can be made independent of the modifications in the CBGT network. This allows the user to derive predictions about network function and behavior in a variety of experimental contexts, which can be vetted against empirical observations. Moreover, various changes in the parameters of the network, as well as the experimental paradigm, can be made through the higher-level `configuration` variable that is sent as an argument in running the simulation, thereby avoiding a considerable coding effort on the part of the user. CBGTPy also returns behavioral outcomes (e.g., choices made and decision times) and "recordings" of neuronal outputs (instantaneous firing rates) for all of the CBGT nuclei in the form of easily usable and readable data frames. Overall, CBGTPy allows for theorists and experimentalists alike to develop and test novel theories of the biological function of these critical pathways.

The individual components of CBGTPy are all designed to enable maximum flexibility. The basal ganglia model is constructed in an organized series of steps, beginning with high-level descriptions of the model and gradually providing more fine-grained details. Developing a modification to the network becomes a matter of inserting or modifying the appropriate components or steps, allowing high-level redesigns to be implemented as easily as more precise low-level modifications. CBGTPy's high degree of extensibility can, in large part, be attributed to its use of a data-flow programming paradigm. Neural pathways between major populations, for example, can be specified at a very high level, requiring only a single entry in the pathway table to describe how each subpopulation is connected. If the connectivity of a particular subpopulation, or even a particular neuron, needs adjustment, then the later steps in network construction can be adjusted to implement those changes. CBGTPy was designed with this degree of flexibility to ensure that in the future, more complex biological models of the CBGT network can be developed and implemented in an efficient manner.

Of course, CBGTPy is not the only neural network model of these cortical-subcortical networks. Many other models exist that describe the circuit-level dynamics of CBGT pathways as either a spiking [24, 37–42] or a rate-based [7, 8, 43–49] system. CBGTPy has some limitations worth noting, such as not being as computationally efficient as rate-based models in generating macroscale dynamics, including those observed using fMRI or EEG, and associated predictions. Also, the properties of the cortical systems modeled in CBGTPy are quite simple and do not capture the nuanced connectivity and representational structure of real cortical systems. For these sorts of questions, there are many other modeling packages that would be better suited for generating hypotheses (e.g., [50]). Where CBGTPy excels is in its a) biologically realistic description of subcortical pathways, b) scalability of adding in new pathways or network properties as they are discovered, c) flexibility at emulating a variety of typical neuroscientific testing environments, and d) ease of use for individuals with relatively limited programming experience. These benefits should make CBGTPy an ideal tool for many researchers interested in basal ganglia and thalamic pathways during behavior.

One issue that has been left unresolved in our toolbox is the problem of parameter fitting [51, 52]. Spiking network models like those used in CBGTPy have an immense number of free parameters. The nature of both the scale and variety of parameters in spiking neural networks makes the fitting problem substantially more complex than that faced by more abstracted neural network models, such as those used in deep learning and modern artificial intelligence [53, 54]. This is particularly true when the goal is to constrain both the neural and behavioral

properties of the network. Models like CBGTPy can be tuned to prioritize matching cellular level properties observed empirically (for example see [15]) or to emphasize matching task performances to humans or non-human participants (see [16]). We view this as a weighted cost function between network dynamics and behavioral performance whose balance depends largely on the goals of the study. To the best of our knowledge, there is no established solution to simultaneously fitting both constraints together in these sorts of networks. Therefore, CBGTPy is designed to be flexible to a wide variety of tuning approaches depending on the goal of the user, rather than constrain to a single fitting method.

Because our focus is on matching neural and behavioral constraints based on experimental observations, CBGTPy's environment was designed to emulate the sorts of task paradigms used in systems and cognitive neuroscience research. We purposefully constructed the environment interface to accommodate a wide variety of traditional and current experimental behavioral tasks. These tasks are often simpler in design than the more complex and naturalistic paradigms used in artificial intelligence and, to an increasing degree, cognitive science. Nonetheless, a long-term goal of CBGTPy development is to interface with environments like OpenAI's Gym [55] in order to provide not only a mechanistic link towards more naturalistic behavior, but also a framework to test hypotheses about the underlying mechanisms of more dynamic and naturalistic behaviors.

In summary, CBGTPy offers a simple way to start generating predictions about CBGT pathways in hypothesis-driven research. This tool enables researchers to run virtual experiments in parallel with *in vivo* experiments in both humans and non-human animals. The extensible nature of the tool makes it easy to introduce updates or expansions in complexity as new observations come to light, positioning it as a potentially important and highly useful tool for understanding these pathways.

## Supporting information

**S1 Appendix. CBGT network.**
(PDF)

**S2 Appendix. Dopamine-dependent plasticity of corticostriatal weights.**
(PDF)

**S3 Appendix. CGBTpy installation and dependencies.**
(PDF)

**S4 Appendix. List of files.**
(PDF)

**S5 Appendix. Network scaling.**
(PDF)

**S1 Table. Relation of all parameters editable by the user.** Here we list all those features that the user can modify and those that cannot. If so, we indicate in which table of the Supplementary information the specific parameters are described.
(PDF)

**S2 Table. Neuronal parameters editable by the user.** These parameters can be modified through the data frame `params`.
(PDF)

**S3 Table. Population-specific neuron parameters changeable by the user.** These parameters can be modified through the dictionary `pops`, addressing the population of interest.
(PDF)

**S4 Table. Synaptic parameters changeable by the user.** These parameters can be modified through the data frame `receps`.
(PDF)

**S5 Table. Population-specific baseline parameters modifiable by the user.** These parameters can be modified through the dictionary `base`, addressing the population of interest.
(PDF)

**S6 Table. Dopamine-related parameters editable by the user.** These parameters can be modified through the data frame `dpmns`.
(PDF)

**S7 Table. Dopamine-related parameters for corticostriatal projections to dSPN and iSPN neurons.** Each of the striatal SPN population, dSPN and iSPN, maintain a copy of this data structure which can be independently modified through the data frames `dSPN_params` and `iSPN_params` defined in the `configuration` dictionary in the notebooks.
(PDF)

**S8 Table. Parameters used for plasticity implementation.** For more details about the plasticity parameters, please refer to S2 Appendix. The parameters without a subscript can be modified using data frame `dpmns`, whereas the parameters with a subscript `dSPN` or `iSPN` can be modified through the data frames `dSPN_params` and `iSPN_params` respectively.
(PDF)

**S9 Table. Parameters that can be set for stop signal stimulation.** The example values included in the table describe the parameters used to generate Fig 7.
(PDF)

**S10 Table. Parameters that can be set for optogenetic stimulation.** The example values included in the table describe the parameters used to generate Fig 8.
(PDF)

**S1 Fig. List of parameters and data frames that are returned from the simulation.**
(TIF)

**S2 Fig. Example of `results['popfreqs']` data frame.**
(TIF)

**S3 Fig. Example of `datatables[0]` data frame.**
(TIF)

**S4 Fig. Example of `Q_df` data frame.**
(TIF)

## Acknowledgments

We would like to thank all the member of exploratory intelligence group, especially Julia Badyna and Dr. Eric Yttri, for their helpful inputs.

## Author Contributions

**Conceptualization:** Matthew Clapp, Jyotika Bahuguna, Cristina Giossi, Jonathan E. Rubin, Timothy Verstynen, Catalina Vich.

**Formal analysis:** Matthew Clapp, Jyotika Bahuguna, Cristina Giossi.

**Funding acquisition:** Jonathan E. Rubin, Timothy Verstynen, Catalina Vich.

**Investigation:** Matthew Clapp, Jyotika Bahuguna, Cristina Giossi.

**Methodology:** Matthew Clapp, Jyotika Bahuguna, Cristina Giossi, Jonathan E. Rubin, Timothy Verstynen, Catalina Vich.

**Project administration:** Jonathan E. Rubin, Timothy Verstynen, Catalina Vich.

**Software:** Matthew Clapp, Jyotika Bahuguna, Cristina Giossi.

**Supervision:** Jonathan E. Rubin, Timothy Verstynen, Catalina Vich.

**Validation:** Matthew Clapp, Jyotika Bahuguna, Cristina Giossi.

**Visualization:** Matthew Clapp, Jyotika Bahuguna, Cristina Giossi, Jonathan E. Rubin, Timothy Verstynen, Catalina Vich.

**Writing – original draft:** Matthew Clapp, Jyotika Bahuguna, Cristina Giossi, Jonathan E. Rubin, Timothy Verstynen, Catalina Vich.

**Writing – review & editing:** Matthew Clapp, Jyotika Bahuguna, Cristina Giossi, Jonathan E. Rubin, Timothy Verstynen, Catalina Vich.

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
