## [Decision Letter · Decision Letter 0]

19 Jun 2024

PONE-D-24-16549CBGTPy: An extensible cortico-basal ganglia-thalamic framework for modeling biological decision makingPLOS ONE

Dear Dr. Vich,

Thank you for submitting your manuscript to PLOS ONE. After careful consideration, we feel that it has merit but does not fully meet PLOS ONE’s publication criteria as it currently stands. Therefore, we invite you to submit a revised version of the manuscript that addresses the points raised during the review process.

Please, improve justification of your model and approach to address questions raised by the reviewers.  How did you calibrate the model parameters? Discuss availability of electrophysiological data for calibration.

Please, provide a detailed description of the model framework including a summary or table that summarizes parameter open for change. What features are set by the design and what features can be changed. Is the number of neurons given in the supplements constant or extra neurons or populations can be added? What model formalisms are used for individual neurons, integrate-and-fire or conductance-based Hodgkin-Huxley? Do the authors use their own integration libraries or are they relying on other platforms, such as the NEURON simulation environment or the NetPyNE platform? Explain the design decisions/choices more thoroughly, and whether one could easily interface agents/environment software with standard modeling platforms mentioned above.

Please, review and consider discussing suggested articles

https://pubmed.ncbi.nlm.nih.gov/36943842/

https://pubmed.ncbi.nlm.nih.gov/32040519/

https://pubmed.ncbi.nlm.nih.gov/28408878/

In introduction, please provide some references for this statement:

"Although there are a wealth of biologically realistic simulations of cortical and non-cortical pathways in the literature, these are often designed to address very narrow behaviors and lack flexibility for testing predictions across multiple experimental contexts."

"We note that the inclusion of a biologically-realistic, dopamine-based learning mechanism, in contrast to the error gradient and backpropagation schemes present in standard artificial agents, represents an important feature of the model in CBGTPy."

Methods:

Is there a reason Ray was chosen rather than the more standard Mpi4py or MPI?

Fig. 3 - are there individual spiking neurons modeled or is this a rate-based model? Provide more details of the neuronal circuits and how they are simulated. Is the NEURON or BRIAN simulator used? Or is a custom-written neuronal simulator used?

Please, clarify which learning algorithm is primarily used in CBGTPy.

Section 3.3 - Why are excitatory/inhibitory currents considered optogenetic rather than electrical or other forms of stimulation?

We look forward to receiving your revised manuscript.

Kind regards,

Gennady S. Cymbalyuk, Ph.D.

Academic Editor

PLOS ONE

 [MC, JB, TV and JER are partly supported by National Institutes of Health (https://www.nih.gov) awards R01DA053014 and R01DA059993 as part of the CRCNS program.  CG and CV are supported by the PCI2020-112026 project, and CV is also supported by the PCI2023-145982-2, both funded by MCIN/AEI/10.13039/501100011033 (https://www.ciencia.gob.es/site/MICINN/aei) and by the European Union "NextGenerationEU"/PRTR (https://next-generation-eu.europa.eu/) as part of the CRCNS program. CG is also supported by the Conselleria de Fons Europeus, Universitat i Cultura del Govern de les Illes Balears (https://www.caib.es/sites/participacio/ca/l/conselleria_de_fons_europeus_universitat_i_cultura/) under grant FPU2023-008-B. The authors assert that these sponsors did not influence the study design, data collection and analysis, decision to publish, or preparation of the manuscript.].  

Additional Editor Comments (if provided):

Reviewers' comments:

Reviewer's Responses to Questions

**Comments to the Author**

1. Is the manuscript technically sound, and do the data support the conclusions?

Reviewer #1: Yes

Reviewer #2: Partly

Reviewer #3: Yes

2. Has the statistical analysis been performed appropriately and rigorously? 

Reviewer #1: N/A

Reviewer #2: N/A

Reviewer #3: N/A

3. Have the authors made all data underlying the findings in their manuscript fully available?

Reviewer #1: Yes

Reviewer #2: Yes

Reviewer #3: Yes

4. Is the manuscript presented in an intelligible fashion and written in standard English?

Reviewer #1: Yes

Reviewer #2: Yes

Reviewer #3: Yes

5. Review Comments to the Author

Reviewer #1: The manuscript “CBGTPy: An extensible cortico-basal ganglia-thalamic framework for modeling biological decision making” presents a biologically relevant, extendable model of the cortico-basal gaglia-thalamo-cortical loop. This model is both tunable to reproduce decision-making tasks and has biological realism to gain insights into the mechanisms of decision making at play.

I appreciate versatility of the model. However, it’s not clear what features are set by the design and what features can be changed. In particular, is the number of neurons given in the supplements constant or extra neurons or populations can be added? I would appreciate a summary or table that summarizes this.

Another problem is how the lack of uniqueness of solution may affect their validity. System reconstruction is an inverse problem that has multiple solutions. In large tunable models, fitting procedures may end up with completely unphysiological or otherwise implausible parameter sets. These models fit better if provided not only behavioral, but also electrophysiological data to restrict the solutions. This is not the case for this model as far as I understand. Another way of modeling is much more restrictive models, where the number of neural groups and connections are minimal (or close to that) for a particular task. A couple of papers I can mention here are actually very relevant to the subject of this manuscript, but not cited (https://pubmed.ncbi.nlm.nih.gov/36943842/
https://pubmed.ncbi.nlm.nih.gov/32040519/
https://pubmed.ncbi.nlm.nih.gov/28408878/ ). However, in the current manuscript, it’s somewhere in the middle. Thus, it poses two problems: First is the lack of unique solution I mention before and/or second, the need to calibrate a vast number of parameters that are not tunable (or plastic). These parameters require data for their calibration, which may not be available. I suggest the authors discuss how these issues can be addressed in this model. The two given examples may serve for this purpose as well: how the parameters were calibrated?

The manuscript is very well written and organized, and I think a minor revision addressing the above comments will suffice.

Reviewer #2: Overall, the authors present and describe their CBGTPy software framework, which allows simulating

the cortico-basal ganglia-thalamic circuits using Python. The authors describe the software

with a lot of detail on how to use it and show several example tasks that CBGTPy supports. Overall,

the tool will help researchers conduct modeling/simulation experiments to test their ideas on

circuit mechanisms contributing to decision making and learning. The higher level of detail

compared to artificial neural networks is a welcome benefit for the computational neuroscience

community to explore.

Although the tool is described adequately from the naive end-user point of view, there are many details

on how the neural network models are constructed that are important for expert computational neuroscientists.

For example, what models are used for individual neurons, integrate-and-fire or conductance-based Hodgkin-Huxley

models? Depending on the answer to that question, further details on the modeling framework are needed. Do

the authors use their own integration libraries or are they relying on other platforms, such as the NEURON

simulation environemnt or the NetPyNE platform? While I commend the authors if they developed their own

platform, of course, that could lead to software fragementation and bugs, many of which may have been

solved already with the more established packages. For that reason, I would ask the authors to

explain their design decisions/choices more thoroughly, and whether they could have simply interfaced

agents/environment software with standard modeling platforms mentioned above.

In addition to the above questions, I wonder whether the authors could further relate their

work to existing, recent models which explored biologically-inspired learning in more complex,

simulated environments. I am curious how difficult would it be to extend the CBGTpy framework to allow simulation of

arbitrary goal-oriented environments with agents embedded within them? The research community would benefit

from being able to interface CBGTPy with software packages such as OpenAI's Gym, where more complex sensory

stimuli and decision making would be interesting to explore. Touching on these considerations

would broaden the appeal of the software. There are in fact some

recent papers that use multi-layer cortical spiking neuronal networks and dopamine-inspired learning

to train agents to play video games and perform other behaviors. Comparison of the present work with

these models would be interesting and provide further context for the author's work:

Training spiking neuronal networks to perform motor control using reinforcement and evolutionary learning, 2022

Training a spiking neuronal network model of visual-motor cortex to play a virtual racket-ball game using reinforcement learning, 2022

Perhaps a naive suggestion, but some of the games mentioned might be cast into the n-choice tasks, since

there are limited choices at every time step of the game, and a reward may be produced based on decision.

However, games have more complex sensory stimuli, which CBGTPy might not be able to handle, unless

it were interfaced with additional neuronal networks that processed the stimuli. If this were not possible,

the downside is of course that more naturalistic style behaviors could not be modeled with the CBGTPy framework.

Detailed comments:

in introduction, please provide some references for this statement:

"Although there are a wealth of

biologically realistic simulations of cortical and non-cortical pathways in the literature,

these are often designed to address very narrow behaviors and lack flexibility for testing

predictions across multiple experimental contexts."

"We note that the inclusion of a biologically-realistic,

dopamine-based learning mechanism, in contrast to the error gradient and

backpropagation schemes present in standard artificial agents, represents an important

feature of the model in CBGTPy."

I agree this is important, and could shed light on how biological systems actually learn.

Methods:

Is there a reaosn Ray was chosen rather than the more standard Mpi4py or MPI?

Fig. 3 - are there individual spiking neurons modeled or is this a rate-based model? I would like

to hear more details of the neuronal circuits and how they are simulated. Is the NEURON or BRIAN

simulator used? Or is a custom-written neuronal simulator used?

Q-learning process - I was under the impression that biologically-inspired learning based on dopamine

system was used. But then the authors talk about Q-learning, which is a more abstract learning algorithm

lacking many biological details/realism. Can the authors clarify which learning algorithm is primarily

used in CBGTPy?

Section 3.3 - Why are excitatory/inhibitory currents considered optogenetic rather than electrical

or other forms of stimulation? It's true that as implemented, they are targeted towards a specific

population, but I would not call it "optogenetic", as it bears little similarity to real

optogenetic mechanisms.

Reviewer #3: The authors have introduced an extensible generative package for modeling cortico-basal ganglia-thalamic pathway (GBGT), called CBGTPy, which simulates neuron responses of various experimental tests, mimicking the functioning of the CBGT of mammalian brain. Although the models are simplifications of the real brain circuit, it allowed the authors to address real experimental questions concerning neural dynamics and behavior. They thus the package in a variety of real life applications, simulating environmental and/or external stimuli, showing the related modifications that these stimuli induced in the CBGT dynamics. The authors carried out an excellent job in describing the package and documenting its parameters. Furthermore, they demonstrated the simplicity in the sintax used for defining these models of brain circuit, which allowed theorists and experimentalists to test novel hypothesis without needing to be expert software developer.

6. PLOS authors have the option to publish the peer review history of their article (what does this mean?). If published, this will include your full peer review and any attached files.

Reviewer #1: No

Reviewer #2: No

Reviewer #3: No

---

## [Author Response · Author response to Decision Letter 0]

5 Aug 2024

We thank all reviewers for their comments and helpful suggestions. We have responded below each of their requests. Changes are highlighted in red on the manuscript.

Reviewer #1: The manuscript “CBGTPy: An extensible cortico-basal ganglia-thalamic framework for modeling biological decision making” presents a biologically relevant, extendable model of the cortico-basal ganglia-thalamo-cortical loop. This model is both tunable to reproduce decision-making tasks and has biological realism to gain insights into the mechanisms of decision making at play.

1. I appreciate versatility of the model. However, it’s not clear what features are set by the design and what features can be changed. In particular, is the number of neurons given in the supplements constant or extra neurons or populations can be added? I would appreciate a summary or table that summarizes this.

Reply: We thank the reviewer for appreciating our model and completely agree with the need for clarity on the modifiable features of the network. We have taken your suggestion of adding a table summarizing the list of model features that can be modified. We have included the Supplementary Table S1 in the manuscript. In addition, we have introduced two new tables indicating the parameters that the user can modify with respect to the stop signal (Supplementary Table S9) and the optogenetic (Supplementary Table S10) experiments.

2. Another problem is how the lack of uniqueness of solution may affect their validity. System reconstruction is an inverse problem that has multiple solutions. In large tunable models, fitting procedures may end up with completely unphysiological or otherwise implausible parameter sets. These models fit better if provided not only behavioral, but also electrophysiological data to restrict the solutions. This is not the case for this model as far as I understand. Another way of modeling is much more restrictive models, where the number of neural groups and connections are minimal (or close to that) for a particular task. A couple of papers I can mention here are actually very relevant to the subject of this manuscript, but not cited (https://pubmed.ncbi.nlm.nih.gov/36943842/
https://pubmed.ncbi.nlm.nih.gov/32040519/
https://pubmed.ncbi.nlm.nih.gov/28408878/ ). However, in the current manuscript, it’s somewhere in the middle. Thus, it poses two problems: First is the lack of unique solution I mention before and/or second, the need to calibrate a vast number of parameters that are not tunable (or plastic). These parameters require data for their calibration, which may not be available. I suggest the authors discuss how these issues can be addressed in this model. The two given examples may serve for this purpose as well: how the parameters were calibrated?

Reply: The reviewer is absolutely correct regarding the problem of parameter identification in these sorts of models. We do not include a built in parameter optimization routine because the goals of parameter tuning/fitting depend largely on the nature of the hypothesis being addressed. For example, if one wanted to use CBGTPy to ask a question about cellular level firing rates or other micro/mesoscale dynamics, then the tuning would be on the cell firing rates itself (see https://journals.plos.org/ploscompbiol/article?id=10.1371/journal.pcbi.1010255 for an example of this type of firing). However, someone interested in fits to macroscopic behavioral responses (e.g., reaction times and choices) might want to relax the constraints on cellular-level properties (see https://elifesciences.org/articles/85223 for an example of this type of fit with CBGTPy). In our experience the simultaneous constraints of fine behavioral tuning *and* cellular property fits is a difficult task in its own right and one that, so far as we know, hasn’t been demonstrated yet. Therefore, we leave the method of parameter tuning to the user. Our prior work includes examples of genetic algorithms and other approaches we have taken in order to identify working parameter sets. 

However, this is likely to be a concern for general readers as well. Therefore, we now include in the Discussion an elaboration of the issue of parameter fitting.

“One issue that has been left unresolved in our toolbox is the problem of parameter fitting \\cite{oyedotun2017simple, abdolrasol2021artificial}. Spiking network models like those used in CBGTPy have an immense number of free parameters. The nature of both the scale and variety of parameters in spiking neural networks makes the fitting problem substantially more complex than that faced by more abstracted neural network models, such as those used in deep learning and modern artificial intelligence \\cite{carlson2014efficient, rossant2011fitting}. This is particularly true when the goal is to constrain both the neural and behavioral properties of the network. Models like CBGTPy can be tuned to prioritize matching cellular level properties observed empirically (for example see \\cite{vich2022identifying}) or to emphasize matching task performances to humans or non-human participants (see \\cite{bond2023competing}). We view this as a weighted cost function between network dynamics and behavioral performance whose balance depends largely on the goals of the study. To the best of our knowledge, there is no established solution to simultaneously fitting both constraints together in these sorts of networks. Therefore, CBGTPy is designed to be flexible to a wide variety of tuning approaches depending on the goal of the user, rather than constrain to a single fitting method.”

However, to help the readers understand the nature of the parameter choices used in our model, we include Table S1_3 in the Appendix, where we have references for the choice of the weights referring to the Arkypallidal pathway simulations and striatal loop as illustration of the way we attempt to constrain the neural dynamics. For the other connections, we redirect the reader to Wei et al 2015, Lo et al 2006, Kumar et al. 2011, Dunovan et al. 2019. We also refer the reader to paper Vich et al. 2022, in which a study is made on the effects of changing these connections and where there is a table with all the references for the choice of the different weights (also added to S1 Appendix as Table S1_5). Then, we have included, in the S1 Appendix, the sentence

“Connections were adjusted to reflect empirical knowledge about local and distal connectivity associated with different populations (see \\cite{wei2015, lo2006cortico, kumar2011role, dunovan2019reward} and the specific connection references in Table S1_3), as well as resting and task-related firing patterns (see Table S1_4 and \\cite{vich2022identifying}). We note that our connection probabilities are generally high (unlike some experimental work such as \\cite{wilson2013active}) due to the fact that we simulate a small number of action channels \\cite{klaus2017spatiotemporal}. In addition, the SPN outputs have been scaled to reflect the fact that the SPN population in our model is relatively small.”

3. The manuscript is very well written and organized, and I think a minor revision addressing the above comments will suffice.

Reply: We thank the reviewer for their helpful comments. Although minor, they do help to improve the accessibility of the work.

Reviewer #2: Overall, the authors present and describe their CBGTPy software framework, which allows simulating the cortico-basal ganglia-thalamic circuits using Python. The authors describe the software with a lot of detail on how to use it and show several example tasks that CBGTPy supports. Overall, the tool will help researchers conduct modeling/simulation experiments to test their ideas on

circuit mechanisms contributing to decision making and learning. The higher level of detail compared to artificial neural networks is a welcome benefit for the computational neuroscience community to explore.

1. Although the tool is described adequately from the naive end-user point of view, there are many details on how the neural network models are constructed that are important for expert computational neuroscientists.

For example, what models are used for individual neurons, integrate-and-fire or conductance-based Hodgkin-Huxley models? Depending on the answer to that question, further details on the modeling framework are needed. 

Reply: The individual neurons are modeled as integrate-and-fire-or-burst (IFB) units (Smith et al., J. Neurophysiol., 2000). The choice to use an integrate-and-fire (IF) style framework was made for computational efficiency, relative to models like the Hodgkin-Huxley model, and to allow us to focus on network connectivity and plasticity, rather than details about contributions of specific ion currents, which would be better studied in a smaller-scale model. The choice to use IFB in particular, rather than a regular IF model, was made to endow the network with bursting capabilities like those observed experimentally, and hence to allow the opportunity for future simulations of pathological states (such as those in a Parkinsonian regime, in which CBGT bursting is elevated). We provide this information in Appendix S1.2, as we mention in the Introduction. In the revision process, we have edited the text at the start of Section 2 to include the following statement, along with a referral to the Appendix: 

“Within each region, we model a collection of spiking point neurons, modeled in a variant of the integrate-and-fire framework [Smith et al., 2000] to include the spiking needed for synaptic plasticity while still maintaining computational efficiency.”

2. Do the authors use their own integration libraries or are they relying on other platforms, such as the NEURON simulation environemnt or the NetPyNE platform? While I commend the authors if they developed their own platform, of course, that could lead to software fragementation and bugs, many of which may have been solved already with the more established packages. For that reason, I would ask the authors to explain their design decisions/choices more thoroughly, and whether they could have simply interfaced agents/environment software with standard modeling platforms mentioned above.

Reply: This is an excellent point. We perform numerical integration in Python, accelerated via Cython, rather than using platforms such as NEURON or BRIAN. The neuron model we are using in CBGTPy is a relatively simple one, a conductance-based integrate-and-fire model, whereas one of the core strengths of NEURON and NetPyNE are in their ability to simulate multiscale models down to the morphological and biophysical levels. For our project it was necessary to weigh the strengths of those platforms against the additional complexity that would be introduced by them, specifically the additional interfacing required to create the network model and manipulate it throughout the course of simulation. Ultimately we selected to use our own numerical integration, as this enables our agent-environment interface to have very direct and simple control over the network simulation. The flexibility of the agent-environment interface was a core aim of the project, more so than detailed and high-precision simulation of neurons, and the resulting numerical integration code represents only a small fraction of the overall codebase.

We now include a brief description of this in the main text, Section 2, as follows:

“Numerical integration is performed via custom Cython code, rather than relying on existing frameworks, such as NEURON \\cite{Carnevale_Hines_2006}, BRIAN \\cite{BRIAN}, or NetPyNE \\cite{NetPyNE}, a design choice which simplified the overall software stack. The core strengths of these frameworks are in the simulation of multi-scale or multi-compartment models, whereas one of the strengths of the CBGTPy model is the high level of direct control that can be exerted over the neural parameters throughout the interactions between the network and its environment (see Section 2.1). The integration is performed in a partially-vectorized manner, in which each variable is represented as a list of Numpy arrays, one array per neural population.”

3. In addition to the above questions, I wonder whether the authors could further relate their

work to existing, recent models which explored biologically-inspired learning in more complex,

simulated environments. I am curious how difficult would it be to extend the CBGTpy framework to allow simulation of arbitrary goal-oriented environments with agents embedded within them? The research community would benefit from being able to interface CBGTPy with software packages such as OpenAI's Gym, where more complex sensory stimuli and decision making would be interesting to explore. Touching on these considerations would broaden the appeal of the software. There are in fact some recent papers that use multi-layer cortical spiking neuronal networks and dopamine-inspired learning to train agents to play video games and perform other behaviors. Comparison of the present work with these models would be interesting and provide further context for the author's work:

Training spiking neuronal networks to perform motor control using reinforcement and evolutionary learning, 2022

Training a spiking neuronal network model of visual-motor cortex to play a virtual racket-ball game using reinforcement learning, 2022

Perhaps a naive suggestion, but some of the games mentioned might be cast into the n-choice tasks, since there are limited choices at every time step of the game, and a reward may be produced based on decision. However, games have more complex sensory stimuli, which CBGTPy might not be able to handle, unless it were interfaced with additional neuronal networks that processed the stimuli. If this were not possible, the downside is of course that more naturalistic style behaviors could not be modeled with the CBGTPy framework.

Reply: We thank the reviewer for the important suggestion. Although the aim of developing CBGTPy has been to develop a modeling framework that simulates experimental paradigms frequently used for understanding cognitive concepts (eg. decision making), an extension to include more naturalistic behavior and interfacing with software packages like OpenAI’s Gym is an ambition that we eventually hope to strive towards. However, apart from the obvious implementation challenges, this also includes several conceptual challenges that will require us to make assumptions that are still currently under exploration. For example, the excellent suggestion made by the reviewer to pose some of the games (say virtual racket ball game) as an n-choice task (which would also be the first approach to be used by the authors), is a concept which still is very much under exploration (https://www.ncbi.nlm.nih.gov/pmc/articles/PMC7815426/). This should first be tested systematically using experimental paradigms on simpler network models with less complex parameter fitting constraints (see our response to Reviewer #1 comment #2). Hence, our approach when designing CBGTPy has been to extend its functionality incrementally by starting with simple paradigms that are more popular in non-human research paradigms, where there is ample published evidence on simultaneous behavioral and neural properties. As those experimental paradigms increase in complexity of behavior we can extend the behavioral properties of CBGTPy. For example, we are currently working on a version of CBGTPy that learns action timing, consistent with studies on continuous vigor control (see https://www.nature.com/articles/nature17639). These sorts of future developments of the network, while exciting, go beyond the goal of the current manuscript. 

Nonetheless, we think that this is an important point for the general reader. As such we have added a paragraph to the Discussion highlighting the future potential of this sort of integration with resources like OpenAI Gym, in order to increase the overall extensibility of the system.

“Because our focus is on matching neural and behavioral constraints based on experimental observations, CBGTPy's environment was designed to emulate the sorts of task paradigms used in systems and cognitive neuroscience research. We purposefully constructed the environment interface t

---

## [Decision Letter · Decision Letter 1]

30 Aug 2024

CBGTPy: An extensible cortico-basal ganglia-thalamic framework for modeling biological decision making

PONE-D-24-16549R1

Dear Dr. Vich,

We’re pleased to inform you that your manuscript has been judged scientifically suitable for publication and will be formally accepted for publication once it meets all outstanding technical requirements.

Kind regards,

Gennady S. Cymbalyuk, Ph.D.

Academic Editor

PLOS ONE

Additional Editor Comments (optional):

Reviewers' comments:

Reviewer's Responses to Questions

**Comments to the Author**

1. If the authors have adequately addressed your comments raised in a previous round of review and you feel that this manuscript is now acceptable for publication, you may indicate that here to bypass the “Comments to the Author” section, enter your conflict of interest statement in the “Confidential to Editor” section, and submit your "Accept" recommendation.

Reviewer #1: All comments have been addressed

Reviewer #2: All comments have been addressed

2. Is the manuscript technically sound, and do the data support the conclusions?

Reviewer #1: Yes

Reviewer #2: Yes

3. Has the statistical analysis been performed appropriately and rigorously? 

Reviewer #1: Yes

Reviewer #2: Yes

4. Have the authors made all data underlying the findings in their manuscript fully available?

Reviewer #1: Yes

Reviewer #2: Yes

5. Is the manuscript presented in an intelligible fashion and written in standard English?

Reviewer #1: Yes

Reviewer #2: Yes

6. Review Comments to the Author

Reviewer #1: All comments has been addressed. I'm glad to recommend it for publication. I'm glad to recommend it for publication.

Reviewer #2: Thanks for addressing my concerns. . . . . . . . . . . . . . . . . . . . . . . . . . . . . I added extra "." character since the character limit is 100 to 20000 Characters, and the minimum bound is enforced.

7. PLOS authors have the option to publish the peer review history of their article (what does this mean?). If published, this will include your full peer review and any attached files.

Reviewer #1: No

Reviewer #2: No

---

## [Editor Report · Acceptance letter]

5 Sep 2024

PONE-D-24-16549R1 

PLOS ONE

Dear Dr. Vich, 

I'm pleased to inform you that your manuscript has been deemed suitable for publication in PLOS ONE. Congratulations! Your manuscript is now being handed over to our production team.

Kind regards, 

on behalf of

Dr. Gennady S. Cymbalyuk 

Academic Editor

PLOS ONE